# Reconstruction for Powerful Graph Representations

**Leonardo Cotta**
Purdue University
cotta@purdue.edu

**Christopher Morris**
Mila – Quebec AI Institute, McGill University
chris@christophermorris.info

**Bruno Ribeiro**
Purdue University
ribeiro@cs.purdue.edu

## Abstract

Graph neural networks (GNNs) have limited expressive power, failing to represent many graph classes correctly. While more expressive graph representation learning (GRL) alternatives can distinguish some of these classes, they are significantly harder to implement, may not scale well, and have not been shown to outperform well-tuned GNNs in real-world tasks. Thus, devising simple, scalable, and expressive GRL architectures that also achieve real-world improvements remains an open challenge. In this work, we show the extent to which graph reconstruction—reconstructing a graph from its subgraphs—can mitigate the theoretical and practical problems currently faced by GRL architectures. First, we leverage graph reconstruction to build two new classes of expressive graph representations. Secondly, we show how graph reconstruction boosts the expressive power of any GNN architecture while being a (provably) powerful inductive bias for invariances to vertex removals. Empirically, we show how reconstruction can boost GNN's expressive power—while maintaining its invariance to permutations of the vertices—by solving seven graph property tasks not solvable by the original GNN. Further, we demonstrate how it boosts state-of-the-art GNN's performance across nine real-world benchmark datasets.

## 1 Introduction

Supervised machine learning for graph-structured data, i.e., graph classification and regression, is ubiquitous across application domains ranging from chemistry and bioinformatics [8, 94] to image [89], and social network analysis [33]. Consequently, machine learning on graphs is an active research area with numerous proposed approaches—notably GNNs [21, 42, 45] being the most representative case of GRL methods.

Arguably, GRL's most interesting results arise from a cross-over between graph theory and representation learning. For instance, the representational limits of GNNs are upper-bounded by a simple heuristic for the graph isomorphism problem [78, 105], the 1-*dimensional Weisfeiler-Leman algorithm* (1-WL) [44, 73, 75, 101, 102], which might miss crucial structural information in the data [5]. Further works show how GNNs cannot approximate graph properties such as diameter, radius, girth, and subgraph counts [25, 38], inspiring architectures [6, 66, 77, 78] based on the more powerful $\kappa$-*dimensional Weisfeiler-Leman algorithm* ($\kappa$-WL) [44].[1] On the other hand, despite the limited expressiveness of GNNs, they still can overfit the training data, offering limited generalization performance [105]. Hence, devising GRL architectures that are simultaneously sufficiently expressive and avoid overfitting remains an open problem.

---

[1] We opt for using $\kappa$ instead of $k$, i.e., $\kappa$-WL instead of $k$-WL, to not confuse the reader with the hyperparameter $k$ of our models.

35th Conference on Neural Information Processing Systems (NeurIPS 2021).

An under-explored connection between graph theory and GRL is graph reconstruction, which studies graphs and graph properties uniquely determined by their subgraphs. In this direction, both the pioneering work of Shawe-Taylor [88] and the more recent work of Bouritsas et al. [18], show that assuming the reconstruction conjecture (see Conjecture 1) holds, their models are most-expressive representations (universal approximators) of graphs. Unfortunately, Shawe-Taylor's computational graph grows exponentially with the number of vertices, and Bouritsas et al.'s full representation power requires performing multiple graph isomorphism tests on potentially large graphs (with $n-1$ vertices). Moreover, these methods were not inspired by the more general subject of graph reconstruction; instead, they rely on the reconstruction conjecture to prove their architecture's expressive powers.

**Contributions.** In this work, we directly connect graph reconstruction to GRL. We first show how the *k-reconstruction of graphs*—reconstruction from induced $k$-vertex subgraphs—induces a natural class of expressive GRL architectures for supervised learning with graphs, denoted *k-Reconstruction Neural Networks*. We then show how several existing works have their expressive power limited by $k$-reconstruction. Further, we show how the reconstruction conjecture's insights lead to a provably most expressive representation of graphs. Unlike Shawe-Taylor [88] and Bouritsas et al. [18], which, for graph tasks, require fixed-size unattributed graphs and multiple (large) graph isomorphism tests, respectively, our method represents bounded-size graphs with vertex attributes and does not rely on isomorphism tests.

To make our models scalable, we propose *k-Reconstruction GNNs*, a general tool for boosting the expressive power and performance of GNNs with graph reconstruction. Theoretically, we characterize their expressive power showing that $k$-Reconstruction GNNs can distinguish graph classes that the 1-WL and 2-WL cannot, such as cycle graphs and strongly regular graphs, respectively. Further, to explain gains in real-world tasks, we show how reconstruction can act as a lower-variance risk estimator when the graph-generating distribution is invariant to vertex removals. Empirically, we show that reconstruction enhances GNNs' expressive power, making them solve multiple synthetic graph property tasks in the literature not solvable by the original GNN. On real-world datasets, we show that the increase in expressive power coupled with the lower-variance risk estimator boosts GNNs' performance up to 25%. Our combined theoretical and empirical results make another important connection between graph theory and GRL.

## 1.1 Related work

We review related work from GNNs, their limitations, data augmentation, and the reconstruction conjecture in the following. See Appendix A for a more detailed discussion.

**GNNs.** Notable instances of this architecture include, e.g., [31, 47, 97], and the spectral approaches proposed in, e.g., [19, 30, 56, 72]—all of which descend from early work in [10, 57, 69, 70, 71, 87, 90]. Aligned with the field's recent rise in popularity, there exists a plethora of surveys on recent advances in GNN methods. Some of the most recent ones include [21, 104, 114].

**Limits of GNNs.** Recently, connections to Weisfeiler-Leman type algorithms have been shown [9, 27, 39, 40, 64, 66, 77, 78, 105]. Specifically, the authors of [78, 105] show how the 1-WL limits the expressive power of any possible GNN architecture. Morris et al. [78] introduce $\kappa$-*dimensional GNNs* which rely on a more expressive message-passing scheme between subgraphs of cardinality $\kappa$. Later, this was refined in [6, 66] and in [76] by deriving models equivalent to the more powerful $\kappa$-dimensional Weisfeiler-Leman algorithm. Chen et al. [27] connect the theory of universal approximation of permutation-invariant functions and graph isomorphism testing, further introducing a variation of the 2-WL. Recently, a large body of work propose enhancements to GNNs, e.g., see [3, 11, 14, 18, 80, 98, 109], making them more powerful than the 1-WL; see [75] for a survey and see Appendix A for a in-depth discussion. For clarity, throughout this work, we will use the term GNNs to denote the class of message-passing architectures limited by the 1-WL algorithm, where the class of distinguishable graphs is well understood [5].

**Data augmentation, generalization and subgraph-based inductive biases.** There exist few works proposing data augmentation for GNNs for graph classification. Kong et al. [59] introduces a simple feature perturbation framework to achieve this, while Rong et al. [85], Feng et al. [34] focus on vertex-level tasks. Garg et al. [38] study the generalization abilities of GNNs showing bounds on the Rademacher complexity, while Liao et al. [62] offer a refined analysis within the PAC-Bayes framework. Recently, Bouritsas et al. [18] proposed to use subgraph counts as vertex and edge features in GNNs.

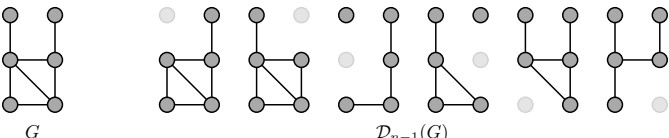

Figure 1: A graph $G$ and its deck $\mathcal{D}_{n-1}(G)$, faded out vertices are not part of each card in the deck.

Although the authors show an increase in expressiveness, the extent, e.g., which graph classes their model can distinguish, is still mostly unclear. Moreover, Yehudai et al. [107] investigate GNNs' ability to generalize to larger graphs. Concurrently, Bevilacqua et al. [12] show how subgraph densities can be used to build size-invariant graph representations. However, the performance of such models in in-distribution tasks, their expressiveness, and scalability remain unclear. Finally, Yuan et al. [111] show how GNNs' decisions can be explained by (often large) subgraphs, further motivating our use of graph reconstruction as a powerful inductive bias for GRL.

**Reconstruction conjecture.** The reconstruction conjecture is a longstanding open problem in graph theory, which has been solved in many particular settings. Such results come in two flavors. Either proving that graphs from a specific class are reconstructible or determining which graph functions are reconstructible. Known results of the former are, for instance, that regular graphs, disconnected graphs, and trees are reconstructible [16, 53]. In particular, we highlight that outerplanar graphs, which account for most molecule graphs, are known to be reconstructible [41]. For a comprehensive review of graph reconstruction results, see Bondy [16].

## 2 Preliminaries

Here, we introduce notation and give an overview of the main results in graph reconstruction theory [16, 43], including the reconstruction conjecture [96], which forms the basis of the models in this work.

**Notation and definitions.** As usual, let $[n] = \{1, \ldots, n\} \subset \mathbb{N}$ for $n \geq 1$, and let $\{\!\{\ldots\}\!\}$ denote a multiset. In an abuse of notation, for a set $X$ with $x$ in $X$, we denote by $X - x$ the set $X \setminus \{x\}$. We also assume elementary definitions from graph theory, such as graphs, directed graphs, vertices, edges, neighbors, trees, isomorphism, et cetera; see Appendix B. The vertex and the edge set of a graph $G$ are denoted by $V(G)$ and $E(G)$, respectively. The *size* of a graph $G$ is equal to its number of vertices. Unless indicated otherwise, we use $n := |V(G)|$. If not otherwise stated, we assume that vertices and edges are annotated with *attributes*, i.e., real-valued vectors.

We denote the set of all finite and simple graphs by $\mathcal{G}$. The subset of $\mathcal{G}$ without edge attributes (or edge directions) is denoted $\mathfrak{G} \subset \mathcal{G}$. We write $G \simeq H$ if the graphs $G$ and $H$ are isomorphic. Further, we denote the isomorphism type, i.e., the equivalence class of the isomorphism relation, of a graph $G$ as $\mathcal{I}(G)$. Let $S \subseteq V(G)$, then $G[S]$ is the induced subgraph with edge set $E(G)[S] = \{S^2 \cap E(G)\}$. We will refer to induced subgraphs simply as subgraphs in this work.

Let $\mathfrak{R}$ be a family of graph representations, such that for $d \geq 1$, $r$ in $\mathfrak{R}$, $r \colon \mathcal{G} \to \mathbb{R}^d$, assigns a $d$-dimensional representation vector $r(G)$ for a graph $G$ in $\mathcal{G}$. We say $\mathfrak{R}$ can *distinguish* a graph $G$ if there exists $r$ in $\mathfrak{R}$ that assigns a unique representation to the isomorphism type of $G$, i.e., $r(G) = r(H)$ if and only if $G \simeq H$. Further, we say $\mathfrak{R}$ distinguishes a pair of non-isomorphic graphs $G$ and $H$ if there exists some $r$ in $\mathfrak{R}$ such that $r(G) \neq r(H)$. Moreover, we write $\mathfrak{R}_1 \preceq \mathfrak{R}_2$ if $\mathfrak{R}_2$ distinguishes between all graphs $\mathfrak{R}_1$ does, and $\mathfrak{R}_1 \equiv \mathfrak{R}_2$ if both directions hold. The corresponding strict relation is denoted by $\prec$. Finally, we say $\mathfrak{R}$ is a *most-expressive representation* of a class of graphs if it distinguishes all non-isomorphic graphs in that class.

**Graph reconstruction.** Intuitively, the reconstruction conjecture states that an undirected edge-unattributed graph can be fully recovered up to its isomorphism type given the multiset of its vertex-deleted subgraphs' isomorphism types. This multiset of subgraphs is usually referred to as the *deck* of the graph, see Figure 1 for an illustration. Formally, for a graph $G$, we define its deck as $\mathcal{D}_{n-1}(G) = \{\!\{\mathcal{I}(G[V(G) - v]) \colon v \in V(G)\}\!\}$. We often call an element in $\mathcal{D}_{n-1}(G)$ a *card*. We define the graph reconstruction problem as follows.

**Definition 1.** *Let $G$ and $H$ be graphs, then $H$ is a* reconstruction *of $G$ if $H$ and $G$ have the same deck, denoted $H \sim G$. A graph $G$ is* reconstructible *if every reconstruction of $G$ is isomorphic to $G$, i.e., $H \sim G$ implies $H \simeq G$.*

Similarly, we define *function reconstruction*, which relates functions that map two graphs to the same value if they have the same deck.

**Definition 2.** *Let $f \colon \mathcal{G} \to \mathcal{Y}$ be a function, then $f$ is* reconstructible *if $f(G) = f(H)$ for all graphs in $\{(H, G) \in \mathcal{G}^2 \colon H \sim G\}$, i.e., $G \sim H$ implies $f(G) = f(H)$.*

We can now state the reconstruction conjecture, which in short says that every $G$ in $\mathfrak{G}$ with $|V| \geq 3$ is reconstructible.

**Conjecture 1** (Kelly [52], Ulam [96]). *Let $H$ and $G$ in $\mathfrak{G}$ be two finite, undirected, simple graphs with at least three vertices. If $H$ is a reconstruction of $G$, then $H$ and $G$ are isomorphic.*

We note here that the reconstruction conjecture does not hold for directed graphs, hypergraphs, and infinite graphs [16, 92, 93]. In particular, edge directions can be seen as edge attributes. Thus, the reconstruction conjecture does not hold for the class $\mathcal{G}$. In contrast, the conjecture has been proved for practical-relevant graph classes, such as disconnected graphs, regular graphs, trees, and outerplanar graphs [16]. Further, computational searches show that graphs with up to eleven vertices are reconstructible [68]. Finally, many graph properties are known to be reconstructible, such as every size subgraph count, degree sequence, number of edges, and the characteristic polynomial [16].

**Graph $k$-reconstruction.** Kelly et al. [53] generalized graph reconstruction, considering the multiset of subgraphs of size $k$ instead of $n - 1$, which we denote $\mathcal{D}_k(G) = \{\!\{ \mathcal{I}(H) \colon H \in \mathcal{S}^{(k)}(G) \}\!\}$, where $\mathcal{S}^{(k)}$ is the set of all $\binom{n}{k}$ $k$-size subsets of $V$. We often call an element in $\mathcal{D}_k(G)$ a $k$-*card*. From the $k$-deck definition, it is easy to extend the concept of graph and function reconstruction, cf. Definitions 1 and 2, to *graph* and *function $k$-reconstruction*.

**Definition 3.** *Let $G$ and $H$ be graphs, then $H$ is a $k$-reconstruction of $G$ if $H$ and $G$ have the same $k$-deck, denoted $H \sim_k G$. A graph $G$ is $k$-reconstructible if every $k$-reconstruction of $G$ is isomorphic to $G$, i.e., $H \sim_k G$ implies $H \simeq G$.*

Accordingly, we define $k$-*function reconstruction* as follows.

**Definition 4.** *Let $f \colon \mathcal{G} \to \mathcal{Y}$ be a function, then $f$ is $k$-reconstructible if $f(G) = f(H)$ for all graphs in $\{(H, G) \in \mathcal{G}^2 \colon H \sim_k G\}$, i.e., $G \sim_k H$ implies $f(G) = f(H)$.*

Results for $k$-reconstruction usually state the least $k$ as a function of $n$ such that all graphs $G$ in $\mathcal{G}$ (or some subset) are $k$-reconstructible [84]. There exist extensive partial results in this direction, mostly describing $k$-reconstructibility (as a function of $n$) for a particular family of graphs, such as trees, disconnected graphs, complete multipartite graphs, and paths, see [84, 60]. More concretely, Nỳdl [83], Spinoza and West [91] showed graphs with $2k$ vertices that are not $k$-reconstructible. In practice, these results imply that for some fixed $k$ there will be graphs with not many more vertices than $k$ that are not $k$-reconstructible. Further, $k$-reconstructible graph functions such as degree sequence and connectedness have been studied in [65, 91] depending on the size of $k$. In Appendix C, we discuss further such results.

## 3 Reconstruction Neural Networks

Building on the previous section, we propose two neural architectures based on graph $k$-reconstruction and graph reconstruction. First, we look at $k$-*Reconstruction Neural Networks*, the most natural way to use graph $k$-reconstruction. Secondly, we look at *Full Reconstruction Neural Networks*, where we leverage the Reconstruction Conjecture to build a most-expressive representation for the class of graphs of bounded size and unattributed edges.

$k$-**Reconstruction Neural Networks.** Intuitively, the key idea of $k$-Reconstruction Neural Networks is that of learning a joint representation based on subgraphs induced by $k$ vertices. Formally, let $f_{\mathbf{W}} \colon \cup_{m=1}^{\infty} \mathbb{R}^{m \times d} \to \mathbb{R}^t$ be a (row-wise) permutation-invariant function and $\mathcal{G}_k = \{G \in \mathcal{G} \colon |V(G)| = k\}$ be the set of graphs with exactly $k$ vertices. Further, let $h^{(k)} \colon \mathcal{G}_k \to \mathbb{R}^{1 \times d}$ be a graph representation function such that two graphs $G$ and $H$ on $k$ vertices are mapped to the same vectorial representation if and only if they are isomorphic, i.e., $h^{(k)}(G) = h^{(k)}(H) \iff G \simeq H$ for all $G$ and $H$ in $\mathcal{G}_k$. We define $k$-Reconstruction Neural Networks over $\mathcal{G}$ as a function with parameters $\mathbf{W}$ in the form

$$r_{\mathbf{W}}^{(k)}(G) = f_{\mathbf{W}} \left( \text{CONCAT}(\{\!\{ h^{(k)}(G[S]) \colon S \in \mathcal{S}^{(k)} \}\!\}) \right),$$

where $\mathcal{S}^{(k)}$ is the set of all $k$-size subsets of $V(G)$ for some $3 \leq k \leq n$, and CONCAT denotes row-wise concatenation of a multi-set of vectors in some arbitrary order. Note that $h^{(k)}$ might also be a function

with learnable parameters. In that case, we require it to be most-expressive for $\mathcal{G}_k$. The following results characterize the expressive power of the above architecture.

**Proposition 1.** *Let $f_{\mathbf{W}}$ be a universal approximator of multisets [112, 100, 79]. Then, $r_{\mathbf{W}}^{(k)}$ can approximate a function if and only if the function is $k$-reconstructible.*

Moreover, we can observe the following.

**Observation 1** (Nỳdl [84], Kostochka and West [60])**.** *For any graph $G$ in $\mathcal{G}$, its $k$-deck $\mathcal{D}_k(G)$ determines its $(k-1)$-deck $\mathcal{D}_{k-1}(G)$.*

From Observation 1, we can derive a hierarchy in the expressive power of $k$-Reconstruction Neural Networks with respect to the subgraph size $k$. That is, $r_{\mathbf{W}}^{(3)} \preceq r_{\mathbf{W}}^{(4)} \preceq \cdots \preceq r_{\mathbf{W}}^{(n-2)} \preceq r_{\mathbf{W}}^{(n-1)}$.

In Appendix D, we show how many existing architectures have their expressive power limited by $k$-reconstruction. We also refer to Appendix D for the proofs, a discussion on the model's computational complexity, approximation methods, and relation to existing work.

**Full Reconstruction Neural Networks.** Here, we propose a recursive scheme based on the reconstruction conjecture to build a most-expressive representation for graphs. Intuitively, Full Reconstruction Neural Networks recursively compute subgraph representations based on smaller subgraph representations. Formally, let $\mathfrak{G}_{\leq n^*}^{\dagger} := \{G \in \mathfrak{G} \colon |V(G)| \leq n^*\}$ be the class of undirected graphs with unattributed edges and maximum size $n^*$. Further, let $f_{\mathbf{W}}^{(k)} \colon \cup_{m=1}^{\infty} \mathbb{R}^{m \times d} \to \mathbb{R}^t$ be a (row-wise) permutation invariant function and let $h_{\{i,j\}}$ be a most-expressive representation of the two-vertex subgraph induced by vertices $i$ and $j$. We can now define the representation $r(G[V(G)])$ of a graph $G$ in $\mathfrak{G}_{\leq n^*}^{\dagger}$ in a recursive fashion as

$$r(G[S]) = \begin{cases} f_{\mathbf{W}}^{(|S|)} \left( \text{CONCAT}(\{\!\{r(G[S-v]) \colon v \in S\}\!\}) \right), & \text{if } 3 \leq |S| \leq n \\ h_S(G[S]), & \text{if } |S| = 2. \end{cases}$$

Again, CONCAT is row-wise concatenation in some arbitrary order. Note that in practice, it is easier to build the subgraph representations in a bottom-up fashion. First, use two-vertex subgraph representations to compute all three-vertex subgraph representations. Then, perform this inductively until we arrive at a single whole-graph representation. In Appendix E, we prove the expressive power of Full Reconstruction Neural Networks, i.e., we show how if the reconstruction conjecture holds, it is a most-expressive representation of undirected edge-unattributed graphs. Finally, we show its quadratic number of parameters, exponential computational complexity, and relation to existing work.

## 4 Reconstruction Graph Neural Networks

Although Full Reconstruction Neural Networks provide a most-expressive representation for undirected, unattributed-edge graphs, they are impractical due to their computational cost. Similarly, $k$-Reconstruction Neural Networks are not scalable since increasing their expressive power requires computing most-expressive representations of larger $k$-size subgraphs. Hence, to circumvent the computational cost, we replace the most-expressive representations of subgraphs from $k$-Reconstruction Neural Networks with GNN representations, resulting in what we name *$k$-Reconstruction GNNs*. This change allows for scaling the model to larger subgraph sizes, such as $n-1, n-2, \ldots$, et cetera.

Since, in the general case, graph reconstruction assumes most-expressive representations of subgraphs, it cannot capture $k$-Reconstruction GNNs' expressive power directly. Hence, we provide a theoretical characterization of the expressive power of $k$-Reconstruction GNNs by coupling graph reconstruction and the GNN expressive power characterization based on the 1-WL algorithm. Nevertheless, in Appendix F.2, we devise conditions under which $k$-Reconstruction GNNs have the same power as $k$-Reconstruction Neural Networks. Finally, we show how graph reconstruction can act as a (provably) powerful inductive bias for invariances to vertex removals, which boosts the performance of GNNs even in tasks where all graphs are already distinguishable by them (see Appendix G). We refer to Appendix F for a discussion on the model's relation to existing work.

Formally, let $f_{\mathbf{W}} \colon \cup_{m=1}^{\infty} \mathbb{R}^{m \times d} \to \mathbb{R}^t$ be a (row-wise) permutation invariant function and $h_{\mathbf{W}}^{\text{GNN}} \colon \mathcal{G} \to \mathbb{R}^{1 \times d}$ a GNN representation. Then, for $3 \leq k < |V(G)|$, a $k$-Reconstruction GNN takes the form

$$r_{\mathbf{W}}^{(k,\text{GNN})}(G) = f_{\mathbf{W}_1} \left( \text{CONCAT}(\{\!\{h_{\mathbf{W}_2}^{\text{GNN}}(G[S]) \colon S \in \mathcal{S}^{(k)}\}\!\}) \right),$$

with parameters $\mathbf{W} = \{\mathbf{W}_1, \mathbf{W}_2\}$, where $\mathcal{S}^{(k)}$ is the set of all $k$-size subsets of $V(G)$, and CONCAT is row-wise concatenation in some arbitrary order.

**Approximating $r_{\mathbf{W}}^{(k,\mathbf{GNN})}$.** By design, $k$-Reconstruction GNNs require computing GNN representations for all $k$-vertex subgraphs, which might not be feasible for large graphs or datasets. To address this, we discuss a direction to circumvent computing all subgraphs, i.e., approximating $r_{\mathbf{W}}^{(k,\mathbf{GNN})}$ by sampling. One possible choice for $f_{\mathbf{W}}$ is Deep Sets [112], which we use for the experiments in Section 5, where the representation is a sum decomposition taking the form $r_{\mathbf{W}}^{(k,\mathbf{GNN})}(G) = \rho_{\mathbf{W}_1}\left(\sum_{S \in \mathcal{S}^{(k)}} \phi_{\mathbf{W}_2}\left(h_{\mathbf{W}_3}^{\mathbf{GNN}}(G[S])\right)\right)$, where $\rho_{\mathbf{W}_1}$ and $\phi_{\mathbf{W}_2}$ are permutation sensitive functions, such as feed-forward networks. We can learn the $k$-Reconstruction GNN model over a training dataset $\mathcal{D}^{(\mathrm{tr})} := \{(G_i, y_i)\}_{i=1}^{N^{(\mathrm{tr})}}$ and a loss function $l$ by minimizing the empirical risk

$$\widehat{\mathcal{R}}_k(\mathcal{D}^{(\mathrm{tr})}; \mathbf{W}_1, \mathbf{W}_2, \mathbf{W}_3) = \frac{1}{N^{\mathrm{tr}}} \sum_{i=1}^{N^{\mathrm{tr}}} l\left(r_{\mathbf{W}}^{(k,\mathbf{GNN})}(G_i), y_i\right). \tag{1}$$

Equation (1) is impractical for all but the smallest graphs, since $r_{\mathbf{W}}^{(k,\mathbf{GNN})}$ is a sum over all $k$-vertex induced subgraphs $\mathcal{S}^{(k)}$ of $G$. Hence, we approximate $r_{\mathbf{W}}^{(k,\mathbf{GNN})}$ using a sample $\mathcal{S}_B^{(k)} \subset \mathcal{S}^{(k)}$ drawn uniformly at random at every gradient step, i.e., $\widehat{r}_{\mathbf{W}}^{(k,\mathbf{GNN})}(G) = \rho_{\mathbf{W}_1}\left(|\mathcal{S}^{(k)}|/|\mathcal{S}_B^{(k)}| \sum_{S \in \mathcal{S}_B^{(k)}} \phi_{\mathbf{W}_2}\left(h_{\mathbf{W}_3}^{(\mathbf{GNN})}(G[S])\right)\right)$. Due to non-linearities in $\rho_{\mathbf{W}_1}$ and $l$, plugging $\widehat{r}_{\mathbf{W}}^{(k,\mathbf{GNN})}$ into Equation (1) does not provide us with an unbiased estimate of $\widehat{\mathcal{R}}_k$. However, if $l(\rho_{\mathbf{W}_1}(a), y)$ is convex in $a$, in expectation, we will be minimizing a proper upper bound of our loss, i.e., $1/N^{\mathrm{tr}} \sum_{i=1}^{N^{\mathrm{tr}}} l\left(r_{\mathbf{W}}^{(k,\mathbf{GNN})}(G_i), y_i\right) \leq 1/N^{\mathrm{tr}} \sum_{i=1}^{N^{\mathrm{tr}}} l\left(\widehat{r}_{\mathbf{W}}^{(k,\mathbf{GNN})}(G_i), y_i\right)$. In practice, many models rely on this approximation and provide scalable and reliable training procedures, cf. [48, 79, 80, 112].

## 4.1 Expressive power

Now, we analyze the expressive power of $k$-Reconstruction GNNs. It is clear that $k$-Reconstruction GNNs $\preceq k$-Reconstruction Neural Networks, however the relationship between $k$-Reconstruction GNNs and GNNs is not that straightforward. At first, one expects that there exists a well-defined hierarchy—such as the one in $k$-Reconstruction Neural Networks (see Observation 1)—between GNNs, $(n-1)$-Reconstruction GNNs, $(n-2)$-Reconstruction GNNs, and so on. However, *there is no such hierarchy*, as we see next.

**Are GNNs more expressive than $k$-Reconstruction GNNs?** It is well-known that GNNs cannot distinguish regular graphs [5, 78]. By leveraging the fact that regular graphs are reconstructible [53], we show that cycles and circular skip link (CSL) graphs—two classes of regular graphs—can indeed be distinguished by $k$-Reconstruction GNNs, implying that $k$-Reconstruction GNNs are not less expressive than GNNs. We start by showing that $k$-Reconstruction GNNs can distinguish the class of cycle graphs.

**Theorem 1** ($k$-Reconstruction GNNs can distinguish cycles). *Let $G \in \mathfrak{G}$ be a cycle graph with $n$ vertices and $k := n - \ell$. An $(n-\ell)$-Reconstruction GNN assigns a unique representation to $G$ if i) $\ell < (1 + o(1))\left(\frac{2 \log n}{\log \log n}\right)^{1/2}$ and ii) $n \geq (\ell - \log \ell + 1)\left(\frac{e + e \log \ell + e + 1}{(\ell - 1) \log \ell - 1}\right) + 1$ hold.*

The following results shows that $k$-Reconstruction GNNs can distinguish the class of CSL graphs.

**Theorem 2** ($k$-Reconstruction GNNs can distinguish CSL graphs). *Let $G, H \in \mathfrak{G}$ be two non-isomorphic circular skip link (CSL) graphs (a class of 4-regular graphs, cf. [23, 80]). Then, $(n-1)$-Reconstruction GNNs can distinguish $G$ and $H$.*

Hence, if the conditions in Theorem 1 hold, GNNs $\npreceq (n - \ell)$-Reconstruction GNNs. Figure 2 (cf. Appendix F) depicts how $k$-Reconstruction GNNs can distinguish a graph that GNNs cannot. The process essentially breaks the local symmetries that make GNNs struggle by removing one (or a few) vertices from the graph. By doing so, we arrive at distinguishable subgraphs. Since we can reconstruct the original graph with its unique subgraph representations, we can identify it. See Appendix F for the complete proofs of Theorems 1 and 2.

**Are GNNs less expressive than $k$-Reconstruction GNNs?** We now show that GNNs can distinguish graphs that $k$-Reconstruction GNNs with small $k$ cannot. We start with Proposition 2 stating that there exist some graphs that GNNs can distinguish which $k$-Reconstruction GNNs with small $k$ cannot.

**Proposition 2.** *GNNs $\npreceq$ $k$-Reconstruction GNNs for $k \leq \lceil n/2 \rceil$.*

On the other hand, the analysis is more interesting for larger subgraph sizes, e.g., $n - 1$, where there are no known examples of (undirected, edge-unattributed) non-reconstructible graphs. There are graphs distinguishable by GNNs with at least one subgraph not distinguishable by them; see Appendix F. However, the analysis is whether the multiset of all subgraphs' representations can distinguish the original graph. Since we could not find any counter-examples, we conjecture that every graph distinguishable by a GNN is also distinguishable by a $k$-Reconstruction GNN with $k = n - 1$ or possibly more generally with any $k$ close enough to $n$. In Appendix F, we state and discuss the conjecture, which we name WL reconstruction conjecture. If true, the conjecture implies GNNs $\prec$ $(n - 1)$-Reconstruction GNNs. Moreover, if we use the original GNN representation together with $k$-Reconstruction GNNs, Theorems 1 and 2 imply that the resulting model is strictly more powerful than the original GNN.

**Are $k$-Reconstruction GNNs less expressive than higher-order ($\kappa$-WL) GNNs?**

Recently a line of work, e.g., [6, 67, 76], explored higher-order GNNs aligning with the $\kappa$-WL hierarchy. Such architectures have, in principle, the same power as the $\kappa$-WL algorithm in distinguishing non-isomorphic graphs. Hence, one might wonder how $k$-Reconstruction GNNs stack up to $\kappa$-WL-based algorithms. The following result shows that pairs of non-isomorphic graphs exist that a $(n - 2)$-Reconstruction GNN can distinguish but the 2-WL cannot.

**Proposition 3.** *Let 2-GNNs be neural architectures with the same expressiveness as the 2-WL algorithm. Then, $(n - 2)$-Reconstruction GNN $\npreceq$ 2-GNNs $\equiv$ 2-WL.*

As a result of Proposition 3, using a $(n - 2)$-Reconstruction GNN representation together with a 2-GNN increases the original 2-GNN's expressive power.

## 4.2 Reconstruction as a powerful extra invariance for general graphs

An essential feature of modern machine learning models is capturing invariances of the problem of interest [63]. It reduces degrees of freedom while allowing for better generalization [13, 63]. GRL is predicated on invariance to vertex permutations, i.e., assigning the same representation to isomorphic graphs. But are there other invariances that could improve the generalization error?

**$k$-reconstruction is an extra invariance.** Let $P(G, Y)$ be the joint probability of observing a graph $G$ with label $Y$. Any $k$-reconstruction-based model, such as $k$-Reconstruction Neural Networks and $k$-Reconstruction GNNs, by definition assumes $P(G, Y)$ to be invariant to the $k$-deck, i.e., $P(G, Y) = P(H, Y)$ if $\mathcal{D}_k(G) = \mathcal{D}_k(H)$. Hence, our neural architectures for $k$-Reconstruction Neural Networks and $k$-Reconstruction GNNs directly define this extra invariance beyond permutation invariance. *How we do know it is an extra invariance and not a consequence of permutation invariance?* It does not hold on directed graphs [93], where permutation invariance still holds.

**Hereditary property variance reduction.** We now show that the invariance imposed by $k$-reconstruction helps in tasks based on *hereditary properties* [17]. A graph property $\mu(G)$ is called hereditary if it is invariant to vertex removals, *i.e.* $\mu(G) = \mu(G[V(G) - v])$ for every $v \in V(G)$ and $G \in \mathcal{G}$. By induction the property is invariant to every size subgraph, i.e., $\mu(G) = \mu(G[S])$ for every $S \in \mathcal{S}^{(k)}, k \in [n]$ where $\mathcal{S}^{(k)}$ is the set of all $k$-size subsets of $V(G)$. Here, the property is invariant to any given subgraph. For example, every subgraph of a planar graph is also planar, every subgraph of an acyclic graph is also acyclic, any subgraph of a $j$-colorable graph is also $j$-colorable. A more practically interesting (weaker) invariance would be invariance to a few vertex removals. Next we define $\delta$-*hereditary properties* (a special case of a $\preceq$-hereditary property). In short, a property is $\delta$-hereditary if it is a hereditary property for graphs with more than $\delta$ vertices.

**Definition 5** ($\delta$-hereditary property). *A graph property $\mu \colon \mathcal{G} \to \mathcal{Y}$ is said to be $\delta$-hereditary if $\mu(G) = \mu(G[V(G) - v]), \forall v \in V(G), G \in \{H \in \mathcal{G} : |V(H)| > \delta\}$. That is, $\mu$ is uniform in $G$ and all subgraphs of $G$ with more than $\delta$ vertices.*

Consider the task of predicting $Y|G := \mu(G)$. Theorem 3 shows that $k$-Reconstruction GNNs is an invariance that reduces the variance of the empirical risk associated with $\delta$-hereditary property tasks. See Appendix F for the proof.

**Theorem 3** (*k-Reconstruction GNNs for variance reduction of δ-hereditary tasks*). *Let $P(G, Y)$ be a δ-hereditary distribution, i.e., $Y := \mu(G)$ where $\mu$ is a δ-hereditary property. Further, let $P(G, Y) = 0$ for all $G \in \mathcal{G}$ with $|V(G)| \geq \delta + \ell$, $\ell > 0$. Then, for k-Reconstruction GNNs taking the form*

$$\rho_{\mathbf{W}_1}\left(1/|\mathcal{S}^{(k)}| \sum_{S \in \mathcal{S}^{(k)}} \phi_{\mathbf{W}_2}\left(h^{GNN}_{\mathbf{W}_3}(G[S])\right)\right), \text{ if } l(\rho_{\mathbf{W}_1}(a), y) \text{ is convex in } a, \text{ we have}$$

$$Var[\widehat{\mathcal{R}}_k] \leq Var[\widehat{\mathcal{R}}_{GNN}],$$

*where $\widehat{\mathcal{R}}_k$ is the empirical risk of k-Reconstruction GNNs with $k := n - \ell$ (cf. Equation (1)) and $\widehat{\mathcal{R}}_{GNN}$ is the empirical risk of GNNs.*

## 5 Experimental Evaluation

In this section, we investigate the benefits of $k$-Reconstruction GNNs against GNN baselines on both synthetic and real-world tasks. Concretely, we address the following questions:

**Q1.** Does the increase in expressive power from reconstruction (cf. Section 4.1) make $k$-Reconstruction GNNs solve graph property tasks not originally solvable by GNNs?

**Q2.** Can reconstruction boost the original GNNs performance on real-world tasks? If so, why?

**Q3.** What is the influence of the subgraph size in both graph property and real-world tasks?

**Synthetic graph property datasets.** For **Q1** and **Q3**, we chose the synthetic graph property tasks in Table 1, for which GNNs are provably incapable to solve due to their limited expressive power [38, 81]. The tasks are CSL [32], where we classify CSL graphs, the cycle detection tasks 4 CYCLES, 6 CYCLES and 8 CYCLES [98] and the multi-task regression from Corso et al. [28], where we want to determine whether a graph is connected, its diameter, and its spectral radius. See Appendix H for datasets statistics.

**Real-world datasets.** To address **Q2** and **Q3**, we evaluated $k$-Reconstruction GNNs on a diverse set of large-scale, standard benchmark instances [49, 74]. Specifically, we used the ZINC (10K) [32], ALCHEMY (10K) [23], OGBG-MOLFREESOLV, OGBG-MOLESOL, and OGBG-MOLLIPO [49] regression datasets. For the case of graph classification, we used OGBG-MOLHIV, OGBG-MOLPCBA, OGBG-TOX21, and OGBG-TOXCAST [49]. See Appendix H for datasets statistics.

**Neural architectures.** We used the GIN [106], GCN [56], and the PNA [28] architectures as GNN baselines. We always replicated the exact architectures from the original paper, building on the respective PyTorch Geometric implementation [35]. For the OGBG regression datasets, we noticed how using a jumping knowledge layer yields better validation and test results for GIN and GCN. Thus we made this small change. For each of these three architectures, we implemented $k$-Reconstruction GNNs for $k$ in $\{n - 1, n - 2, n - 3, \lceil n/2 \rceil\}$ using a Deep Sets function [112] over the *exact same original GNN architecture*. For more details, see Appendix G.

**Experimental setup.** To establish fair comparisons, we retain all hyperparameters and training procedures from the original GNNs to train the corresponding $k$-Reconstruction GNNs. Tables 1 and 2 and Table 6 in Appendix I present results with the same number of runs as previous work [28, 32, 49, 77, 98], i.e., five for all datasets execpt the OGBG datasets, where we use ten runs. For more details, such as the number of subgraphs sampled for each $k$-Reconstruction GNN and each dataset, see Appendix G.

**Non-GNN baselines.** For the graph property tasks, original work used vertex identifiers or Laplacian embeddings to make GNNs solve them. This trick is effective for the tasks but violates an important premise of graph representations, invariance to vertex permutations. To illustrate this line of work, we compare against Positional GIN, which uses Laplacian embeddings [32] for the CSL task and vertex identifiers for the others [98, 28]. To compare against other methods that like $k$-Reconstruction GNNs are invariant to vertex permutations and increase the expressive power of GNNs, we compare against Ring-GNNs [27] and (3-WL) PPGNs [66]. For real-world tasks, Table 6 in Appendix I shows the results from GRL alternatives that incorporate higher-order representations in different ways, LRP [27], GSN [18], δ-2-LGNN [77], and SMP [98].

All results are fully reproducible from the source and are available at `https://github.com/PurdueMINDS/reconstruction-gnns`.

**Results and discussion.**

**A1 (Graph property tasks).** Table 1 confirms Theorem 2, where the increase in expressive power from reconstruction allows $k$-Reconstruction GNNs to distinguish CSL graphs, a task that GNNs cannot solve. Here, $k$-Reconstruction GNNs boost the accuracy of standard GNNs between $10\times$ and $20\times$. Theorem 2 only guarantees GNN expressiveness boosting for $(n - 1)$-Reconstruction, but our empirical results also show benefits for $k$-Reconstruction with $k \leq n-2$. Table 1 also confirms Theorem 1, where

Table 1: Synthetic graph property tasks. We highlight in green $k$-Reconstruction GNNs boosting the original GNN architecture. [†]: Std. not reported in original work. [+]: Laplacian embeddings used as positional features. [*]: Vertex identifiers used as positional features.

| | CSL (Accuracy % %) ↑ | 4 CYCLES (Accuracy %) ↑ | 6 CYCLES (Accuracy %) ↑ | 8 CYCLES (Accuracy %) ↑ | Multi-task CONNECTIVITY (log MSE) ↓ | DIAMETER (log MSE) ↓ | SPECTRAL RADIUS (log MSE) ↓ | Invariant to vertex permutations? |
|---|---|---|---|---|---|---|---|---|
| **GIN** (orig.) | 4.66 ± 4.00 | 93.0[†] | 92.7[†] | 92.5[†] | -3.419 ± 0.320 | 0.588 ± 0.354 | -2.130 ± 1.396 | ✔ |
| Reconstr. $(n-1)$ | 88.66 ± 22.66 | 95.17 ± 4.91 | 97.35 ± 0.74 | 94.69 ± 2.34 | -3.575 ± 0.395 | -0.195 ± 0.714 | -2.732 ± 0.793 | ✔ |
| $(n-2)$ | 78.66 ± 22.17 | 94.06 ± 5.10 | 97.50 ± 0.72 | 95.04 ± 2.69 | -3.799 ± 0.187 | -0.207 ± 0.381 | -2.344 ± 0.569 | ✔ |
| $(n-3)$ | 73.33 ± 16.19 | 96.61 ± 1.40 | 97.84 ± 1.37 | 94.48 ± 2.13 | -3.779 ± 0.064 | 0.105 ± 0.225 | -1.908 ± 0.860 | ✔ |
| $\lceil n/2 \rceil$ | 40.66 ± 9.04 | 75.13 ± 0.26 | 63.28 ± 0.59 | 63.53 ± 1.14 | -3.765 ± 0.083 | 0.564 ± 0.025 | -2.130 ± 0.166 | ✔ |
| **GCN** (orig.) | 6.66 ± 2.10 | 98.336 ± 0.24 | 95.73 ± 2.72 | 87.14 ± 12.73 | -3.781 ± 0.075 | 0.087 ± 0.186 | -2.204 ± 0.362 | ✔ |
| Reconstr. $(n-1)$ | **100.00** ± 0.00 | 99.00 ± 0.10 | 97.63 ± 0.19 | 94.99 ± 2.31 | **-4.039** ± 0.101 | -1.175 ± 0.425 | -3.625 ± 0.536 | ✔ |
| $(n-2)$ | **100.00** ± 0.00 | 98.77 ± 0.61 | 97.89 ± 0.69 | 97.82 ± 1.10 | -3.970 ± 0.059 | -0.577 ± 0.135 | -3.397 ± 0.273 | ✔ |
| $(n-3)$ | 96.00 ± 6.46 | 99.11 ± 0.19 | 98.31 ± 0.52 | 97.18 ± 0.58 | -3.995 ± 0.031 | -0.333 ± 0.117 | -3.105 ± 0.286 | ✔ |
| $\lceil n/2 \rceil$ | 49.33 ± 7.42 | 75.19 ± 0.19 | 66.04 ± 0.59 | 63.66 ± 0.51 | -3.693 ± 0.063 | 0.8518 ± 0.016 | -1.838 ± 0.054 | ✔ |
| **PNA** (orig.) | 10.00 ± 2.98 | 81.59 ± 19.86 | 95.57 ± 0.36 | 84.81 ± 16.48 | -3.794 ± 0.155 | -0.605 ± 0.097 | -3.610 ± 0.137 | ✔ |
| Reconstr. $(n-1)$ | **100.00** ± 0.00 | 97.88 ± 2.19 | 99.18 ± 0.20 | 98.92 ± 0.72 | **-3.904** ± 0.001 | -0.765 ± 0.032 | **-3.954** ± 0.118 | ✔ |
| $(n-2)$ | 95.33 ± 7.77 | **99.12** ± 0.28 | 99.10 ± 0.57 | **99.22** ± 0.27 | -3.781 ± 0.085 | -0.090 ± 0.135 | -3.478 ± 0.206 | ✔ |
| $(n-3)$ | 95.33 ± 5.81 | 89.36 ± 0.22 | 99.34 ± 0.26 | 93.92 ± 8.15 | -3.710 ± 0.209 | 0.042 ± 0.047 | -3.311 ± 0.067 | ✔ |
| $\lceil n/2 \rceil$ | 42.66 ± 11.03 | 75.34 ± 0.18 | 65.58 ± 0.95 | 64.01 ± 0.30 | -2.977 ± 0.065 | 1.445 ± 0.037 | -1.073 ± 0.075 | ✔ |
| Positional GIN | 99.33[+] ± 1.33 | 88.3[†] | 96.1[†] | 95.3[†] | -1.61[†] | **-2.17**[†] | -2.66[†] | ✗ |
| Ring-GNN | 10.00 ± 0.00 | 99.9[†] | **100.0**[†] | 71.4[†] | — | — | — | ✔ |
| PPGN (3-WL) | 97.80 ± 10.91 | 99.8[†] | 87.1[†] | 76.5[†] | — | — | — | ✔ |

Table 2: OGBG molecule graph classification and regression tasks. We highlight in green $k$-Reconstruction GNNs boosting the original GNN architecture.

| | OGBG-MOLTOX21 (ROC-AUC %) ↑ | OGBG-MOLTOXCAST (ROC-AUC %) ↑ | OGBG-MOLFREESOLV (RSMSE) ↓ | OGBG-MOLESOL (RSMSE) ↓ | OGBG-MOLLIPO (RSMSE) ↓ | OGBG-MOLPCBA (AP %) ↑ |
|---|---|---|---|---|---|---|
| **GIN** (orig.) | 74.91 ± 0.51 | 63.41 ± 0.74 | 2.411 ± 0.123 | 1.111 ± 0.038 | 0.754 ± 0.010 | 21.16 ± 0.28 |
| Reconstr. $(n-1)$ | 75.15 ± 1.40 | 63.95 ± 0.53 | 2.283 ± 0.279 | 1.026 ± 0.033 | **0.716** ± 0.020 | 23.60 ± 0.02 |
| $(n-2)$ | **76.84** ± 0.62 | **65.36** ± 0.49 | **2.117** ± 0.181 | **1.006** ± 0.030 | 0.736 ± 0.025 | 23.25 ± 0.00 |
| $(n-3)$ | 76.78 ± 0.64 | 64.84 ± 0.71 | 2.370 ± 0.326 | 1.055 ± 0.031 | 0.738 ± 0.018 | 23.33 ± 0.09 |
| $\lceil n/2 \rceil$ | 74.40 ± 0.75 | 62.29 ± 0.28 | 2.531 ± 0.206 | 1.343 ± 0.053 | 0.842 ± 0.020 | 13.50 ± 0.32 |
| **GCN** (orig.) | 75.29 ± 0.69 | 63.54 ± 0.42 | 2.417 ± 0.178 | 1.106 ± 0.036 | 0.793 ± 0.040 | 20.20 ± 0.24 |
| Reconstr. $(n-1)$ | 76.46 ± 0.77 | 64.51 ± 0.60 | 2.524 ± 0.300 | 1.096 ± 0.045 | 0.760 ± 0.015 | 21.25 ± 0.25 |
| $(n-2)$ | 75.58 ± 0.99 | 64.38 ± 0.39 | 2.467 ± 0.231 | 1.086 ± 0.048 | 0.766 ± 0.025 | 20.10 ± 0.08 |
| $(n-3)$ | 75.88 ± 0.73 | 64.70 ± 0.81 | 2.345 ± 0.261 | 1.114 ± 0.047 | 0.754 ± 0.021 | 19.04 ± 0.03 |
| $\lceil n/2 \rceil$ | 74.03 ± 0.63 | 62.80 ± 0.77 | 2.599 ± 0.161 | 1.372 ± 0.048 | 0.835 ± 0.020 | 11.69 ± 1.41 |
| **PNA** (orig.) | 74.28 ± 0.52 | 62.69 ± 0.63 | 2.192 ± 0.125 | 1.140 ± 0.032 | 0.759 ± 0.017 | 25.45 ± 0.04 |
| Reconstr. $(n-1)$ | 73.64 ± 0.74 | 64.14 ± 0.76 | 2.341 ± 0.070 | 1.723 ± 0.145 | 0.743 ± 0.015 | 23.11 ± 0.05 |
| $(n-2)$ | 74.89 ± 0.29 | 65.22 ± 0.47 | 2.298 ± 0.115 | 1.392 ± 0.272 | 0.794 ± 0.065 | 22.10 ± 0.03 |
| $(n-3)$ | 75.10 ± 0.73 | 65.03 ± 0.58 | 2.133 ± 0.086 | 1.360 ± 0.163 | 0.785 ± 0.041 | 20.05 ± 0.15 |
| $\lceil n/2 \rceil$ | 73.71 ± 0.61 | 61.25 ± 0.49 | 2.185 ± 0.231 | 1.157 ± 0.056 | 0.843 ± 0.018 | 12.33 ± 1.20 |

$k$-Reconstruction GNNs provide significant accuracy boosts on all cycle detection tasks (4 CYCLES, 6 CYCLES and 8 CYCLES). See Appendix J.1, for a detailed discussion on results for CONNECTIVITY, DIAMETER, and SPECTRAL RADIUS, which also show boostings.

**A2 (Real-world tasks).** Table 2 and Table 6 in Appendix I show that applying $k$-reconstruction to GNNs significantly boosts their performance across all eight real-world tasks. In particular, in Table 2, we see a boost of up to 5% while achieving the best results in five out of six datasets. The $(n-2)$-reconstruction applied to GIN gives the best results in the OGBG tasks, with the exception of OGBG-MOLLIPO and OGBG-MOLPCBA where $(n-1)$-reconstruction performs better. The only settings where we did not get any boost were PNA for OGBG-MOLESOL and OGBG-MOLPCBA. Table 6 in Appendix I also shows consistent boost in GNNs' performance of up to 25% in other datasets. On ZINC, $k$-Reconstruction yields better results than the higher-order alternatives LRP and $\delta$-2-LGNN. While GSN gives the best ZINC results, we note that GSN requires application-specific features. In OGBG-MOLHIV, $k$-reconstruction is able to boost both GIN and GCN. The results in Appendix G show that nearly 100% of the graphs in our real-world datasets are distinguishable by the 1-WL algorithm, thus we can conclude that traditional GNNs are expressive enough for all our real-world tasks. Hence, real-world boosts of reconstruction over GNNs can be attributed to the gains from invariances to vertex removals (cf. Section 4.2) rather than the boost in expressive power (cf. Section 4.1).

**A3 (Subgraph sizes).** Overall we observe that removing one vertex ($k=n-1$) is enough to improve the performance of GNNs in most experiments. At the other extreme end of vertex removals, $k=\lceil n/2 \rceil$, there is a significant loss in expressiveness compared to the original GNN. In most real-world tasks, Table 2 and Table 6 in Appendix I show a variety of performance boosts also with $k \in \{n-2, n-3\}$. For GCN and PNA in OGBG-MOLESOL, specifically, we only see $k$-Reconstruction boosts over smaller subgraphs such as $n-3$, which might be due to the task's need of more invariance to vertex removals (cf.

Section 4.2). In the graph property tasks (Table 1), we see significant boosts also for $k \in \{n-2, n-3\}$ in all models across most tasks, except PNA. However, as in real-world tasks the extreme case of small subgraphs $k = \lceil n/2 \rceil$ significantly harms the ability to solve tasks with $k$-Reconstruction GNNs.

# 6 Conclusions

Our work connected graph ($k$-)reconstruction and modern GRL. We first showed how such connection results in two natural expressive graph representation classes. To make our models practical, we combined insights from graph reconstruction and GNNs, resulting in $k$-Reconstruction GNNs. Our theory shows that reconstruction boosts the expressiveness of GNNs and has a lower-variance risk estimator in distributions invariant to vertex removals. Empirically, we showed how the theoretical gains of $k$-Reconstruction GNNs translate into practice, solving graph property tasks not originally solvable by GNNs and boosting their performance on real-world tasks.

# Acknowledgements

This work was funded in part by the National Science Foundation (NSF) awards CAREER IIS-1943364 and CCF-1918483. Any opinions, findings and conclusions or recommendations expressed in this material are those of the authors and do not necessarily reflect the views of the sponsors. Christopher Morris is funded by the German Academic Exchange Service (DAAD) through a DAAD IFI postdoctoral scholarship (57515245). We want to thank our reviewers, who gave excellent suggestions to improve the paper.

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
