# A Related work (expanded)

**GNNs.** Recently, graph neural networks [42, 87] emerged as the most prominent (supervised) GRL architectures. Notable instances of this architecture include, e.g., [31, 47, 97], and the spectral approaches proposed in, e.g., [19, 30, 56, 72]—all of which descend from early work in [57, 69, 87, 90]. Recent extensions and improvements to the GNN framework include approaches to incorporate different local structures (around subgraphs), e.g., [2, 36, 50, 82, 106], novel techniques for pooling vertex representations in order perform graph classification, e.g., [20, 37, 108, 113], incorporating distance information [110], and non-euclidian geometry approaches [22]. Moreover, recently empirical studies on neighborhood aggregation functions for continuous vertex features [28], edge-based GNNs leveraging physical knowledge [4, 58], and sparsification methods [85] emerged. A survey of recent advancements in GNN techniques can be found, e.g., in [21, 103, 114].

**Limits of GNNs.** Chen et al. [25] study the substructure counting abilities of GNNs. Dasoulas et al. [29], Abboud et al. [1] investigate the connection between random coloring and universality. Recent works have extended GNNs' expressive power by encoding vertex identifiers [80, 98], adding random features [86], using higher-order topology as features [18], considering simplicial complexes [3, 14], encoding ego-networks [109], and encoding distance information [61]. Although these works increase the expressiveness of GNNs, their generalization abilities are understood to a lesser extent. Further, works such as Vignac et al. [98, Lemma 6] and the most recent Beaini et al. [11] and Bodnar et al. [14] prove the boost in expressiveness with a single pair of graphs, giving no insights into the extent of their expressive power or their generalization abilities. For clarity, throughout this work, we use the term GNNs to denote the class of message-passing architectures limited by the 1-WL algorithm, where the class of distinguishable graphs is well understood [5].

# B Notation (expanded)

As usual, let $[n] = \{1, \dots, n\} \subset \mathbb{N}$ for $n \geq 1$, and let $\{\!\{\dots\}\!\}$ denote a multiset. In an abuse of notation, for a set $X$ with $x$ in $X$, we denote by $X - x$ the set $X \setminus \{x\}$.

**Graphs.** A *graph* $G$ is a pair $(V, E)$ with a *finite* set of *vertices* $V$ and a set of *edges* $E \subseteq \{\{u, v\} \subseteq V \mid u \neq v\}$. We denote the set of vertices and the set of edges of $G$ by $V(G)$ and $E(G)$, respectively. For ease of notation, we denote the edge $\{u, v\}$ in $E(G)$ by $(u, v)$ or $(v, u)$. In the case of *directed graphs* $E \subseteq \{(u, v) \in V \times V \mid u \neq v\}$. An *attributed graph* $G$ is a triple $(V, E, \alpha)$ with an attribute function $\alpha \colon V(G) \cup E(G) \to \mathbb{R}^a$ for $a > 0$. Then $\alpha(v)$ is an *attribute* of $v$ for $v$ in $V(G) \cup E(G)$. The *neighborhood* of $v$ in $V(G)$ is denoted by $N(v) = \{u \in V(G) \mid (v, u) \in E(G)\}$. Unless indicated otherwise, we use $n := |V(G)|$.

We say that two graphs $G$ and $H$ are isomorphic, $G \simeq H$, if there exists an adjacency preserving bijection $\varphi \colon V(G) \to V(H)$, i.e., $(u, v)$ is in $E(G)$ if and only if $(\varphi(u), \varphi(v))$ is in $E(H)$, and call $\varphi$ an *isomorphism* from $G$ to $H$. If the graphs have vertex or edge attributes, the isomorphism is additionally required to match these attributes accordingly.

We denote the set of all finite and simple graphs by $\mathcal{G}$. The subset of $\mathcal{G}$ without edge attributes is denoted $\mathfrak{G} \subset \mathcal{G}$. Further, we denote the isomorphism type, i.e., the equivalence class of the isomorphism relation, of a graph $G$ as $\mathcal{I}(G)$. Let $S \subseteq V(G)$, then $G[S]$ is the induced subgraph with edge set $E(G)[S] = \{S^2 \cap E(G)\}$. We will refer to induced subgraphs simply as subgraphs in this work.

# C More on reconstruction

After formulating the Reconstruction Conjecture, it is natural to wonder whether it stands for other relational structures, such as directed graphs. Interestingly, directed graphs, hypergraphs, and infinite graphs are not reconstructible [16, 92]. Thus, in particular the Reconstruction Conjecture does not hold for the class $\mathcal{G}$.

Another question is how many cards from the deck are sufficient to reconstruct a graph. Bollobás [15] show that almost every graph, in a probabilistic sense, can be reconstructed with only three subgraphs from the deck. For example, the graph shown in Figure 1 (Section 2) is reconstructible from the three leftmost cards.

For an extensive survey on reconstruction, we refer the reader to Bondy [16], Godsil [43]. From there, we highlight a significant result, Kelly's Lemma (cf. Lemma 1). In short, the lemma states that the deck of a graph completely defines its subgraph count of every size.

**Lemma 1** (Kelly's Lemma [53])**.** *Let $\nu(H, G)$ be the number of copies of $H$ in $G$. For any pair of graphs $G, H \in \mathcal{G}$ with $V(G) > V(H)$, $\nu(H, G)$ is reconstructible.*

In fact, its proof is very simple once we realize every subgraph $H$ appears in exactly $(|V(G)| - |V(H)|)$ cards from the deck, i.e.,

$$\nu(G, H) = \sum_{v \in V(G)} \frac{\nu(G[V(G) - v], H)}{(|V(G)| - |V(H)|)}.$$

Manvel [65] started the study of graph reconstruction with the $k$-deck, which has been recently reviewed by Kostochka and West [60] and Nỳdl [84]. Related work also refers to $k$-reconstruction as $\ell$-reconstruction [60], where $\ell = n - k$ is the number of deleted vertices from the original graph.

Here, in Lemma 2, we highlight a generalization of Kelly's Lemma (Lemma 1) established in Nỳdl [84], where the count of any subgraph of size at most $k$ is $k$-reconstructible.

**Lemma 2** (Nỳdl [84]). *For any pair of graphs $G, H \in \mathcal{G}$ with $V(G) > k \geq V(H)$, $\nu(H, G)$ is $k$-reconstructible.*

# D  More on $k$-Reconstruction Neural Networks

Here, we give more background on $k$-Reconstruction Neural Networks.

## D.1  Properties

We start by showing how the $k$-ary Relational Pooling framework [80] is a specific case of $k$-Reconstruction Neural Networks and thus limited by $k$-reconstruction. Then, we show how $k$-Reconstruction Neural Networks are limited by $k$-GNNs at initialization, which implies that $k$-GNNs [78] at initialization can approximate any $k$-reconstructible function.

**Observation 2** ($k$-ary Relational Pooling $\preceq$ $k$-Reconstruction Neural Networks). *The $k$-ary pooling approach in the Relational Pooling (RP) framework [80] defines a graph representation of the form*

$$h_{\mathbf{W}}^{(RP)}(G) = \frac{1}{\binom{n}{k}} \sum_{S \in \mathcal{S}^{(k)}} \overrightarrow{h}_{\mathbf{W}}^{(k)}(G[S]),$$

*where $\mathcal{S}^{(k)}$ is the set of all $\binom{n}{k}$ $k$-size subsets of $V$ and $\overrightarrow{h}_{\mathbf{W}}^{(k)}(\cdot)$ is a most-expressive graph representation given by the average of a permutation-sensitive universal approximator, e.g., a feed-forward neural network, applied over the $k!$ permutations of the subgraph, accordingly. Thus, $k$-ary RP can be casted as a $k$-Reconstruction Neural Network with $f_{\mathbf{W}}$ as mean pooling and $h^{(k)}$ as $\overrightarrow{h}^{(k)}$. Note that for $k$-ary RP to be as expressive as $k$-Reconstruction Neural Networks, i.e., $k$-ary RP $\equiv$ $k$-Reconstruction Neural Networks, we would need to replace the average pooling by a universal multiset approximator or simply add a feed-forward neural network after it.*

**Observation 3** ($k$-Reconstruction Neural Networks $\preceq$ $k$-WL at initialization). *The $k$-WL test, which limits architectures such as Morris et al. [77, 78], Maron et al. [66], at initialization, with zero iteration, considers one-hot encodings of the isomorphism type of subgraphs induced by $k$-tuples of vertices. Note that each $k$-size subgraph is completely defined by its corresponding $k!$ vertex tuples. Thus, it follows that $k$-WL with zero iterations is at least as expressive as $k$-Reconstruction Neural Networks. Further, by combining Proposition 1 and the result from Lemma 2 [84], it follows that $k$-WL [77] at initialization can count subgraphs of size $\leq k$, which is a simple proof for the recent result [26, Theorem 3.7].*

Now, we discuss the computational complexity of $k$-Reconstruction Neural Networks and how to circumvent it through subgraph sampling.

**Computational complexity.** As outlined in Section 2, we would need subgraphs of size almost $n$ to have a most-expressive representation of graphs with $k$-Reconstruction Neural Networks. This would imply performing isomorphism testing for arbitrarily large graphs, as in Bouritsas et al. [18], making the model computationally infeasible.

A graph with $n$ vertices has $\binom{n}{k}$ induced subgraphs of size $k$. Let $\mathcal{T}_{h^{(k)}}$ be an upper-bound on computing $h^{(k)}$. Thus, computing $r_{\mathbf{W}}^{(k)}(G)$ would take $\mathcal{O}(\binom{n}{k}\mathcal{T}_{h^{(k)}})$ time. Although Babai [7] has shown how to do isomorphism testing in quasi-polynomial time, an efficient (polynomial) time algorithm remains unknown. More generally, expressive representations of graphs [54, 80] and isomorphism class hashing algorithms [51] still require exponential time regarding the graph size. Thus, if we choose a small value for $k$, i.e., $n \gg k$, the $\binom{n}{k}$ factor dominates, while if we choose $k \approx n$ the $\mathcal{T}_{h^{(k)}}$ factor dominates. In both cases, the time complexity is exponential in $k$, i.e., $\mathcal{O}(n^k)$.

## D.2  Relation to previous work

Recently, Bouritsas et al. [18] propose using subgraph isomorphism type counts as features of vertices and edges used in a GNN architecture. The authors comment that if the reconstruction conjecture holds, their architecture is most expressive for $k = n - 1$. Here, we point out two things. First, their architecture is at least as powerful as $k$-reconstruction. Secondly, the reconstruction conjecture does not hold for directed graphs. Since edge directions can be seen as edge attributes, their architecture is not the most expressive for graphs with attributed edges. Finally, to make their architecture scalable, in practice, the authors choose only specific hand-engineered subgraph types, which makes the model incomparable to $k$-reconstruction.

### D.3 Proof of Proposition 1

We start by giving a more formal statement of Proposition 1.

Let $f$ be a continuous function over a compact set of $\mathcal{G}$ and $||\cdot||$ the uniform (sup) norm. Proposition 1 states that for every $\epsilon > 0$ there exists some $\mathbf{W}_\epsilon$ such that $||f(G) - r^{(k)}_{\mathbf{W}_\epsilon}(G)|| < \epsilon$ if and only if $f$ is $k$-reconstructible.

*Proof.* Since $h^{(k)}$ is required to be most expressive, we can see the input of $k$-Reconstruction Neural Networks as a multiset of unique identifiers of isomorphism types. Thus, it follows from Definition 4, that $k$-reconstrucible functions can be approximated by $r^{(k)}_{\mathbf{W}}$. The other direction, i.e., a function can be approximated by $h^{(k)}$ if it is $k$-reconstructible, follows from the Stone–Weierstrass theorem, see Zaheer et al. [112]. $\qquad\square$

## E  More on Full Reconstruction Neural Networks

The following result captures the expressive power of Full Reconstruction Neural Networks.

**Proposition 4.** *If the functions $f^k_{\mathbf{W}}$ for all $k = 3, ..., n^*$ are universal approximators of multisets [79, 100, 112] and the Reconstruction Conjecture holds, Full Reconstruction Neural Networks can approximate a function if the function is reconstructible.*

*Proof.* We use induction on $|S|$ to show that every subgraph representation in Full Reconstruction Neural Networks is a most expressive representation if the Reconstruction Conjecture holds.

  i) Base case: $|S| = 2$. It follows from the model definition that $r(G[S])$ is a most expressive representation if $|S| = 2$.

  ii) Inductive step: $2 < |S| < n^*$. If all subgraph representations in $\{\!\{r(G[S-v]) \,|\, v \in S\}\!\}$ are most expressive, it follows from Proposition 1 that if $f^{(|S|)}_{\mathbf{W}}$ is a universal approximator of multisets $r(G[S])$ can approximate any reconstructible function. Thus, if the Reconstruction Conjecture holds, $r(G[S])$ can assign a most expressive representation to $G[S]$.

It follows then that $r(G[V(G)])$ will be a multiset function $f^{(n^*)}_{\mathbf{W}}$ of $\{\!\{r(G[V(G) - v]) \,|\, v \in V(G)\}\!\}$. From Proposition 1, if $f^{(n^*)}_{\mathbf{W}}$ is a universal approximator of multisets $r(G[V(G)])$ can approximate any reconstructible function. $\qquad\square$

It follows from Proposition 4 that if the Reconstruction Conjecture holds, Full Reconstruction Neural Networks are a most-expressive representation of $\mathfrak{G}^\dagger_{\leq n^*}$.

**Number of parameters.** Wagstaff et al. [100] shows how a multiset model needs at least $N$ neurons to learn over multisets of size at most $N$. Since our graphs have at most $n^*$ vertices, we can bound the multiset input size of each $\mathbf{f}_{\mathbf{W}_k}$ for all $k = 3, ..., n^*$. Thus, our total number of parameters is $\mathcal{O}(n^{*^2})$.

**Computational complexity.** For a graph with $n$ vertices, we need to compute representations of all subgraphs of sizes $2, 3, \ldots, n$, i.e., $\binom{n}{2} + \binom{n}{3} + \cdots + \binom{n}{n-1}$. Thus, computing a Full Reconstruction Neural Network representation takes $\mathcal{O}(2^{n^*})$ time.

**Relation to previous work.** Unlike Shawe-Taylor [88], the first work proposing reconstruction to build symmetric neural networks for unattributed graphs with a fixed size, we can handle graphs with vertex attributes and of varying sizes of size up to $n^*$. Future work can explore approximate computing methods a Full Reconstruction Neural Network representation, as recently done for the Relational Pooling (RP) framework. Further, in contrast to the most expressive representation in the RP framework, which uses a permutation-sensitive function, Full Reconstruction Neural Networks incorporate graph invariances in the model.

## F  More on $k$-Reconstruction GNNs

In the following, we give more details on $k$-Reconstruction GNNs.

### F.1  Relation to previous work

Recently, Garg et al. [38] showed how many graph properties are not recognizable by GNNs using specific cycle graph examples. Further, most existing work extending GNN architectures to make them more expressive, such as Vignac et al. [98], Li et al. [61], and Murphy et al. [80] focus on distinguishing regular graph examples. Finally,

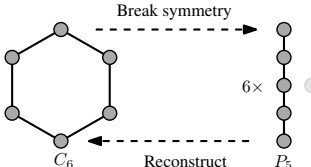

Figure 2: A cycle, undistinguishable by GNNs, and how reconstruction makes it distinguishable.

Beaini et al. [11] used a single pair of planar, non-regular graphs, also used in Garg et al. [38], to prove their method is more expressive than GNNs. Here, we show how the simple idea of graph reconstruction—without changing the original GNN architecture—can extend the GNN's expressivity, distinguishing some classes of regular graphs.

## F.2 Relating $k$-Reconstruction GNNs and $k$-Reconstruction Neural Networks

Here, we devise conditions under which $k$-Reconstruction GNNs and $k$-Reconstruction Neural Networks have the same power, using the following definition.

**Definition 6.** *Let $\mathcal{D}$ be a distribution on graphs. A graph representation is $\varepsilon$-universal with $\varepsilon$ in $[0, 1)$ for $\mathcal{D}$ if it assigns, with probability $(1 - \varepsilon)$, a unique representation, up to isomorphism, to a graph sampled from $\mathcal{D}$. If a graph representation is $\varepsilon$-universal for all induced $k$-vertex subgraphs of graphs sampled from $\mathcal{D}$, then the representation is $(\varepsilon, k)$-universal for $\mathcal{D}$.*

Based on the above definition, we get the following result, relating $k$-Reconstruction GNNs and $k$-Reconstruction Neural Networks.

**Proposition 5.** *Let $\mathcal{D}$ be a distribution on graphs with at most $n^*$ vertices, let $r_{\mathbf{W}}^{(k)}$ be a $k$-Reconstruction Neural Network, let $r_{\mathbf{W}}^{(k,GNN)}$ be a $k$-Reconstruction GNN, and let $h_{\mathbf{W}}^{GNN}$ be the underlying GNN graph representation used within the $k$-Reconstruction GNN $r_{\mathbf{W}}^{(k,GNN)}$. Assume that the GNN $h_{\mathbf{W}}^{GNN}$ is $(\varepsilon/\binom{n^*}{k}, k)$-universal, then with probability $1 - \varepsilon$ it holds that*

$$r_{\mathbf{W}}^{(k,GNN)} \equiv r_{\mathbf{W}}^{(k)}.$$

*Proof.* By the union bound, we can upper bound the probability that at least one $k$-vertex subgraph is not uniquely represented by the GNN $h_{\mathbf{W}}^{GNN}$ by

$$\sum_{i=1}^{\binom{n^*}{k}} \frac{\varepsilon}{\binom{n^*}{k}} = \varepsilon.$$

Hence, we can lower bound the probability that this never happens by $1 - \varepsilon$. □

## F.3 Proof of Theorem 1

In the following, we proof Theorem 1. The following result, showing that the 1-WL assigns unique representation to forest graphs, follows directly from [5].

**Lemma 3** (Arvind et al. [5]). *1-WL distinguishes any pair of non-isomorphic forests.*

The following results shows that the degree list of a graph is reconstructable.

**Lemma 4** (Taylor [95]). *The degree list of an $n$-vertex graph is $(n - \ell)$-reconstructible if*

$$n \geq (\ell - \log \ell + 1)\Big(\frac{e + e\log \ell + e + 1}{(\ell - 1)\log \ell - 1}\Big) + 1.$$

The following results shows that connectedness of a graph is reconstructable.

**Lemma 5** (Spinoza and West [91]). *Connectedness of an $n$-vertex graph is $(n - \ell)$-reconstructible if*

$$\ell < (1 + o(1))\Big(\frac{2\log n}{\log\log n}\Big)^{1/2}.$$

.

Moreover, we will need the following observation.

**Observation 4.** *Every subgraph of a cycle graph is either a path graph or a collection of path graphs.*

We can now prove Theorem 1.

*Proof of Theorem 1.* We start with a simple observation in Observation 4. With that, we know from Lemma 3 that GNNs can assign unique representations to every cycle subgraph. Thus, it follows from Proposition 1 that $k$-Reconstruction GNNs can learn $k$-reconstructible functions of cycle graphs. With that, if conditions *i)* and *ii)* hold, it follows from Lemmas 4 and 5 that we can reconstruct the degree list and the connectedness of a cycle graph. Note that a cycle graph is a 2-regular connected graph. That is, it is uniquely identified by its degree list and its connectedness. Thus, $k$-Reconstruction GNNs can assign an unique representation to it if conditions *i)* and *ii)* hold. $\square$

## F.4 Proof of Theorem 2

The following definition defines CSL graphs.

**Definition 7** (Circular Skip Link (CSL) graphs [80]). *Let $R$ and $M$ be co-prime natural numbers such that $R < M - 1$. We denote by $\mathcal{G}_{skip}(M, R)$ the undirected 4-regular graph with vertices labeled as $0, 1, ..., M - 1$ whose edges form a cycle and have skip links. More specifically, the edge set is defined by a cycle formed by $(i, i + 1), (i + 1, i)$ in $E$ for $i \in \{1, ..., M - 2\}$ and $(0, M - 1), (M - 1, 0)$ together with skip links defined recursively by the sequence of edges $(s_i, s_{i+1}), (s_{i+1}, s_i) \in E$ with $s_1 = 0, s_{i+1} = (s_i + R) \mod M$*

*Proof of Theorem 2.* Consider two non-isomorphic CSL graphs with the same number of vertices, which we can denote by $\mathcal{G}_{skip}(M, R)$ and $\mathcal{G}_{skip}(M, R')$ with $R \neq R'$ according to Definition 7. First, note that every $n - 1$-size subgraph (card) of a CSL graph is isomorphic to each other. Thus, for $k$-Reconstruction GNNs—due to the equivalence in expressiveness between GNNs and 1-WL—it suffices to prove that 1-WL can distinguish between a card from $\mathcal{G}_{skip}(M, R)$ and a card from $\mathcal{G}_{skip}(M, R')$.

Now, let $\mathcal{G}_{skip}^{-i}(M, R)$ and $\mathcal{G}_{skip}^{-i}(M, R')$ be the two subgraphs we get by removing the vertex $i$ from $\mathcal{G}_{skip}(M, R)$ and $\mathcal{G}_{skip}(M, R')$ respectively. In each subgraph, $M - 4$ vertices remain with degree 4. However, we can differentiate the two subgraphs by looking at the vertices which now have degree 3. In both subgraphs $i - 1$ and $i + 1$ will have degree 3. Moreover, in $\mathcal{G}_{skip}^{-i}(M, R)$ the vertex $(i + R) \mod M$ will have degree 3, while $(i + R')$ mod $M$ in $\mathcal{G}_{skip}^{-i}(M, R')$ will have degree 3. Since $(i + R) \mod M \neq (i + R') \mod M$, the distance from $i + 1$ to $(i + R) \mod M$ and to $(i + R') \mod M$ is different in the two graphs. Thus, the 1-WL will assign different colors to the vertices between $i + 1$ and $(i + R) \mod M$ in a subgraph and between $i + 1$ and $(i + R')$ mod $M$ in the other. The same argument applies to the distance from $j$, where $(j + R) \mod M \equiv i$ to $i - 1$. Hence, the 1-WL will assign different color histograms to the subgraphs and thus $k$-Reconstruction GNNs can distinguish them. $\square$

## F.5 Proof of Proposition 2

We start by stating the following result.

**Lemma 6** (Nỳdl [83]). *Spider graphs are not $\lceil n/2 \rceil$-reconstructible.*

*Proof of Proposition 2.* First, it is clear that $k$-Reconstruction GNNs $\preceq$ $k$-Reconstruction Neural Networks, thus it suffices to show GNNs $\npreceq$ $k$-Reconstruction Neural Networks for $k \leq \lceil n/2 \rceil$. It follows from Lemma 6 and Proposition 1 that $k$-Reconstruction Neural Networks cannot assign unique representations to spider graphs if $k = \lceil n/2 \rceil$. However, spider graphs are a family of trees, which are known to be assigned unique representations in 1-WL, see Lemma 3. Thus, due to the equivalence in expressiveness between GNNs and 1-WL, we know GNNs can assign unique representations to spider graphs. Thus, GNNs $\npreceq$ $k$-Reconstruction Neural Networks for $k = \lceil n/2 \rceil$. In fact, from Observation 1, we know GNNs $\npreceq$ $k$-Reconstruction Neural Networks for $k \leq \lceil n/2 \rceil$. $\square$

## F.6 Proof of Proposition 3

In the following, we provide a pair of non-isomorphic graphs that the 2-WL cannot distinguish, while a $k$-Reconstruction GNN with $k := n - 2$ can.

*Proof sketch of Proposition 3.* The 2-WL cannot distinguish any pair of non-isomorphic, strongly-regular graphs with the same parameters [46]. For example, following [46], the 2-WL cannot distinguish the line graph of $K_{4,4}$ (graph $G_1$) and Shrikande Graph (graph $G_2$), both strongly-regular graphs with parameters $(16, 6, 2, 2)$, which are non-isomorphic.[2] Hence, any 2-GNN architecture can also not distinguish them. The graph $G_1$ and $G_2$ are non-isomorphic since the neighborhood around each node in the graph either induces a cycle or a disjoint union of triangles, respectively. By a similar argument as in the proof of Theorem 2, the 1-WL can distinguish the graphs induced by the decks $\mathcal{D}_2(G_1)$ and $\mathcal{D}_2(G_2)$, which we verified by a computer experiment. Hence, there exists a $(n - 2)$-Reconstruction 2-GNN architecture that can distinguish the two graphs. $\square$

---

[2]`https://www.win.tue.nl/~aeb/graphs/srg/srgtab.html`

## F.7 WL reconstruction conjecture

There exists a wide variety of graphs identifiable by 1-WL—and thus by GNNs—with one or a few $(n-1)$-size subgraphs not identifiable by 1-WL. A simple example would be adding a special vertex to a cycle with 5 vertices. In this new 6-vertex graph we connect the special vertex to every other vertex in the 5-cycle. Further, we add another 5-cycle as a different component. This new 11-vertex graph is identifiable by 1-WL from Kiefer et al. [55, Theorem 17].[3] However, the 10-vertex subgraph we get by removing the special vertex is a regular graph, notably not identifiable by 1-WL.

As we saw, getting one or even a few subgraphs that GNNs cannot distinguish from a distinguishable original graph is not a complex task. However, in order to understand whether GNNs are not less powerful than $k$-Reconstruction GNNs we need to find a counter example where 1-WL cannot distinguish the entire multiset of $k$-vertex subgraphs. In this work, we were not able to find such example for large enough $k$, *i.e.* $k \approx n$. Thus, we next state what we name the WL reconstruction conjecture.

**Conjecture 2.** *For $\mathcal{S}^{(k)}$ as the set of all $k$-size subsets of $V(G)$, let $G_k = \sum_{S \in \mathcal{S}^{(k)}} G[S]$ be the disjoint union of all $k$-vertex subgraphs of $G$. Then, there exists some $k \in [n]$ such that if $G$ is uniquely identifiable by 1-WL, $G_k$ is.*

If Conjecture 2 holds, GNNs $\prec$ $k$-Reconstruction GNNs.

## F.8 Proof of Theorem 3

We start by extending the Invariance Lemma [24, Lemma 4.1] to our context in Lemma 7.

**Lemma 7.** *Let $\mu$ be an arbitrary $\delta$-hereditary property and $P_\mathcal{D}$ as in Theorem 3. now, let $\mu_k(G) := \mathbb{E}_{S \sim Unif(\mathcal{S}^{(k)})}[\mu(G[S])] = \frac{1}{|\mathcal{S}^{(k)}|} \sum_{S \in \mathcal{S}^{(k)}} \mu(G[S])$. Then:*

i) *By inspection, for any $G \in \mathcal{G}$ with $|V(G)| \geq \delta + \ell$, $\mu_k(G) = \mathbb{E}[\mu(H) \colon H \in \mathcal{G}_k(G)]$ where $\mathcal{G}_k(G) := \{G[S] \colon S \in \mathcal{S}^{(k)}\}$.*

ii) *By the law of total expectation, $\mathbb{E}_{P_\mathcal{D}}[\mu(G)] = \mathbb{E}_{P_\mathcal{D}}[\mu_k(G)]$.*

iii) *Note that the covariance matrices of $\mu(G)$ and any of its subgraphs $\mu(G[S])$ are equal, i.e., $Cov_{P_\mathcal{D},Unif(\mathcal{S}^{(k)})}\mu(G[S])$. Thus, by the law of total covariance,*

$$Cov_{P_\mathcal{D}}[\mu(G)] = Cov_{P_\mathcal{D}}[\mu_k(G)] + \mathbb{E}_{P_\mathcal{D}}[Cov_{Unif(\mathcal{S}^{(k)})}[\mu(G)]].$$

*Proof of Theorem 3.* Let us first define three risk estimators:

a) GNN estimator:

$$\widehat{\mathcal{R}}_{\mathrm{GNN}}(\mathcal{D}^{(\mathrm{tr})}; \mathbf{W}_2, \mathbf{W}_3) := \frac{1}{N^{\mathrm{tr}}} \sum_{i=1}^{N^{\mathrm{tr}}} l\left(\rho_{\mathbf{W}_1}\left(\phi_{\mathbf{W}_2}\left(h_{\mathbf{W}_3}^{\mathrm{GNN}}(G[S])\right)\right), y_i\right)$$

b) Data augmentation estimator:

$$\widehat{\mathcal{R}}_\circ(\mathcal{D}^{(\mathrm{tr})}; \mathbf{W}_2, \mathbf{W}_3) := \frac{1}{N^{\mathrm{tr}}} \sum_{i=1}^{N^{\mathrm{tr}}} 1/|\mathcal{S}^{(k)}| \sum_{S \in \mathcal{S}^{(k)}} l\left(\rho_{\mathbf{W}_1}\left(\phi_{\mathbf{W}_2}\left(h_{\mathbf{W}_3}^{\mathrm{GNN}}(G[S])\right)\right), y_i\right)$$

c) $k$-Reconstruction GNN estimator:

$$\widehat{\mathcal{R}}_k(\mathcal{D}^{(\mathrm{tr})}; \mathbf{W}_1, \mathbf{W}_2, \mathbf{W}_3) := \frac{1}{N^{\mathrm{tr}}} \sum_{i=1}^{N^{\mathrm{tr}}} l\left(\rho_{\mathbf{W}_1}\left(1/|\mathcal{S}^{(k)}| \sum_{S \in \mathcal{S}^{(k)}} \phi_{\mathbf{W}_2}\left(h_{\mathbf{W}_3}^{\mathrm{GNN}}(G[S])\right)\right), y_i\right)$$

Now, we leverage Lemma 7. By mapping $\phi_{\mathbf{W}_2}\left(h_{\mathbf{W}_3}^{\mathrm{GNN}}(G[S])\right)$ to $\mu$ and $1/|\mathcal{S}^{(k)}| \sum_{S \in \mathcal{S}^{(k)}} l\left(\phi_{\mathbf{W}_2}\left(h_{\mathbf{W}_3}^{\mathrm{GNN}}(G[S])\right), y_i\right)$ to $\mu_k$, from iii) we get that

$$\mathrm{Var}[\widehat{\mathcal{R}}_\circ(\mathcal{D}^{(\mathrm{tr})}; \mathbf{W}_2, \mathbf{W}_3)] \leq \mathrm{Var}[\widehat{\mathcal{R}}_{\mathrm{GNN}}(\mathcal{D}^{(\mathrm{tr})}; \mathbf{W}_2, \mathbf{W}_3)]$$

Since $l \circ \rho_{\mathbf{W}_1}$ is convex in the first argument, we apply Jensen's inequality and arrive at

---

[3]The flip of this graph is a bouquet forest with two 5-cycles (of distinct colors) and an isolated extra vertex

$$\mathrm{Var}[\widehat{\mathcal{R}}_k(\mathcal{D}^{(\mathrm{tr})}; \mathbf{W}_1, \mathbf{W}_2, \mathbf{W}_3)] \leq \mathrm{Var}[\widehat{\mathcal{R}}_\circ(\mathcal{D}^{(\mathrm{tr})}; \mathbf{W}_2, \mathbf{W}_3)] \leq \mathrm{Var}[\widehat{\mathcal{R}}_{\mathrm{GNN}}(\mathcal{D}^{(\mathrm{tr})}; \mathbf{W}_2, \mathbf{W}_3)],$$

as we wanted to show. $\square$

## G  Details on Experiments and Architectures

In the following, we give details on the experiments.

### G.1  1-WL test on real-world datasets.

To show how the expressiveness of GNNs is not an obstacle in real-world tasks, we tested if the 1-WL can distinguish each pair of non-isomorphic graphs in every dataset used in Section 5. We go further and ignore vertex and edge features in the test, showing how only the graphs' topology is enough for GNNs to distinguish $\approx 100\%$ of the graphs in each dataset. Results are shown in Table 3.

Table 3: Percentage of distinguished non-isomorphic graph pairs by 1-WL over the used benchmark datasets.

| Dataset | % of dist. non-iso. graph pairs |
|---|---|
| ZINC | 100.00 % |
| ALCHEMY | >99.99 % |
| OGBG-MOLTOX21 | >99.99 % |
| OGBG-MOLTOXCAST | >99.99 % |
| OGBG-MOLFREESOLV | 100.00 % |
| OGBG-MOLESOL | 100.00 % |
| OGBG-MOLLIPO | 100.00 % |
| OGBG-MOLHIV | >99.99 % |
| OGBG-MOLPCBA | >99.99 % |

### G.2  Architectures.

In the following, we outline details on the used GNN architectures.

**GIN.** Below we specify the architecture together with its $k$-Reconstruction GNN version for each dataset.

ZINC: We used the exact same architecture as used in Morris et al. [77]. Its reconstruction versions used a Deep Sets function with mean pooling with three hidden layers before the pooling and two after it. All hidden layers are of the same size as the GNN layers.

ALCHEMY: We used the exact same architecture as used in Morris et al. [77]. Its reconstruction versions used a Deep Sets function with mean pooling with one hidden layer before the pooling and three after it. All hidden layers are of the same size as the GNN layers.

OGBG-MOLHIV: We used the exact same architecture as used in Hu et al. [49]. Its reconstruction versions used a Deep Sets function with mean pooling with no hidden layer before the pooling and two after it. Additionally, a dropout layer before the output layer. All hidden layers are of the same size as the GNN layers.

OGBG-MOLTOX21: Same as OGBG-MOLHIV.

OGBG-MOLTOXCAST: We used the exact same architecture as used in Hu et al. [49]. Its reconstruction versions used a Deep Sets function with mean pooling with no hidden layer before or after it.

OGBG-MOLFREESOLV: We used the exact same architecture as used in Hu et al. [49], with the exception of using their jumping knowledge layer, which yielded better validation and test results. Its reconstruction versions used a Deep Sets function with mean pooling with no hidden layer before or after it.

OGBG-MOLESOL: Same as OGBG-MOLFREESOLV.

OGBG-MOLLIPO: We used the exact same architecture as used in Hu et al. [49], with the exception of using their jumping knowledge layer, which yielded better validation and test results. Its reconstruction versions used a Deep Sets function with mean pooling with no hidden layer before and three after it. All hidden layers are of the same size as the GNN layers.

OGBG-MOLPCBA: We used the exact same architecture as used in Hu et al. [49]. Its reconstruction versions used a Deep Sets function with mean pooling with one hidden layer before the pooling and no after it. All hidden layers are of the same size as the GNN layers.

CSL: We used the exact same architecture as used in Dwivedi et al. [32]. Its reconstruction versions used a Deep Sets function with mean pooling with no hidden layer before the pooling and one after it. All hidden layers are of the same size as the GNN layers (110 neurons).

MULTITASK: Same as CSL, but with hidden layers of size 300.

4,6,8 CYCLES: We used the same GIN architecture from CSL, with the difference of using one hidden layer and two after the aggregation in the Deep Sets architecture. Additionally, a dropout layer before the output layer. All hidden layers are of the same size as the GNN layers, *i.e.* 300.

**GCN.** We used the exact same architectures from GIN for each dataset with the only change being the convolution (aggregation) layer, here we used the GCN layer from Kipf and Welling [56] instead of GIN.

**PNA.** Below we specify the architecture together with its $k$-Reconstruction GNN version for each dataset.

ZINC: We used the exact same architecture from Corso et al. [28]. Its reconstruction versions used a Deep Sets function with mean pooling with one hidden layer before the pooling and three after it. All hidden layers have 25 hidden units.

ALCHEMY: We used the exact same architecture used for ZINC in Corso et al. [28]. Its reconstruction versions used a Deep Sets function with mean pooling with one hidden layer before the pooling and three after it. All hidden layers have 25 hidden units.

OGBG-MOLHIV: We used the exact same architecture as used in Corso et al. [28]. Its reconstruction versions used a Deep Sets function with mean pooling with one hidden layer before the pooling and three after it. In the original PNA model the hidden layers after vertex pooling are of sizes 70, 35 and 17. We replaced them so all have 70 hidden units and put the layers after the subgraph pooling with sizes 70, 35 and 17.

OGBG-MOLTOX21: Same as OGBG-MOLHIV.

OGBG-MOLTOXCAST: Same as OGBG-MOLHIV.

OGBG-MOLFREESOLV: We used the exact same architecture used for OGBG-MOLHIV in Corso et al. [28]. Its reconstruction versions used a Deep Sets function with mean pooling with no hidden layer before or after the pooling.

OGBG-MOLESOL: Same as OGBG-MOLFREESOLV.

OGBG-MOLLIPO: Same as OGBG-MOLHIV.

OGBG-MOLPCBA: We used the exact same architecture used for OGBG-MOLHIV in Corso et al. [28]. Its reconstruction versions used a Deep Sets function with mean pooling with no hidden layer before and two after the pooling. Additionally, a dropout layer before the output layer. All hidden units are of size 510, with exception of the two hidden layers in Deep Sets that had 255 and 127 neurons.

CSL: Same as ALCHEMY.

MULTITASK: We used the exact same architecture from in Corso et al. [28]. Its reconstruction versions used a Deep Sets function with mean pooling with no hidden layer before and two after the pooling. All hidden units are of size 16.

4,6,8 CYCLES: We used the exact same architecture from in Corso et al. [28], with the difference of a sum pooling instead of a Set2Set [99] pooling for the readout function in PNA. Its reconstruction versions used a Deep Sets function with mean pooling with no hidden layer before and two after the pooling. Additionally, a dropout layer before the output layer. All hidden units are of size 16.

### G.3 Experimental setup.

As mentioned in Section 5, we retain training procedures and evaluation metrics from the original GNN works [32, 77, 49]. We highlight how CSL is the only dataset with a $k$-fold cross validation (with $k = 5$) as originally proposed in [32]. In Table 4, we highlight the number of subgraph samples used for training and testing each $k$-Reconstruction GNN architecture for every dataset, our only new hyperparameter introduced. Note that for test, for what is not specified in Table 4 we use 200 samples or compute exactly (if number of subgraphs $\leq$ 200) for all architectures in all datasets..

**Implementation.** All models were implemented in PyTorch Geometric[35] using NVidia GeForce 1080 Ti GPUs.

## H   Datasets

In Table 5, we show some basic statistics from the datasets used in Section 5.

Table 4: The number of subgraph samples used in training in each $k$-Reconstruction GNN for every dataset. $+$: train/test (validation same as test).

| | | # of samples used in $k$-Reconstruction GNN | | | |
| Dataset | GNN architecture | $n-1$ | $n-2$ | $n-3$ | $\lceil n/2 \rceil$ |
|---|---|---|---|---|---|
| ZINC | GIN | Exact | 10 | 10 | 10 |
| ZINC | GCN | Exact | 10 | 10 | 10 |
| ZINC | PNA | Exact | 10 | 10 | 10 |
| ALCHEMY | GIN | Exact | 30 | 30 | 30 |
| ALCHEMY | GCN | Exact | 30 | 30 | 30 |
| ALCHEMY | PNA | Exact | 30 | 30 | 30 |
| OGBG-MOLTOX21 | GIN | 5 | 5 | 5 | 5 |
| OGBG-MOLTOX21 | GCN | 5 | 5 | 5 | 5 |
| OGBG-MOLTOX21 | PNA | 5 | 5 | 5 | 5 |
| OGBG-MOLTOXCAST | GIN | 5 | 5 | 5 | 5 |
| OGBG-MOLTOXCAST | GCN | 5 | 5 | 5 | 5 |
| OGBG-MOLTOXCAST | PNA | 5 | 5 | 5 | 5 |
| OGBG-MOLFREESOLV | GIN | Exact | Exact | Exact | Exact |
| OGBG-MOLFREESOLV | GCN | Exact | Exact | Exact | Exact |
| OGBG-MOLFREESOLV | PNA | Exact | 20 | 20 | 20 |
| OGBG-MOLESOL | GIN | Exact | Exact | Exact | Exact |
| OGBG-MOLESOL | GCN | Exact | Exact | Exact | Exact |
| OGBG-MOLESOL | PNA | Exact | 20 | 20 | 20 |
| OGBG-MOLLIPO | GIN | Exact | Exact | Exact | Exact |
| OGBG-MOLLIPO | GCN | Exact | Exact | Exact | Exact |
| OGBG-MOLLIPO | PNA | 20 | 20 | 20 | 20 |
| OGBG-MOLHIV | GIN | 5 | 5 | 5 | 5 |
| OGBG-MOLHIV | GCN | 5 | 5 | 5 | 5 |
| OGBG-MOLHIV | PNA | 5 | 5 | 5 | 5 |
| OGBG-MOLPCBA | GIN | $3/5^+$ | $3/5^+$ | $3/5^+$ | $3/5^+$ |
| OGBG-MOLPCBA | GIN | $3/5^+$ | $3/5^+$ | $3/5^+$ | $3/5^+$ |
| OGBG-MOLPCBA | PNA | $1/3^+$ | $1/3^+$ | $1/3^+$ | $1/3^+$ |
| CSL | GIN | Exact | 20 | 20 | 20 |
| CSL | GCN | Exact | 20 | 20 | 20 |
| CSL | PNA | Exact | 20 | 20 | 20 |
| MULTITASK | GIN | $25/20^+$ | $25/20^+$ | $25/20^+$ | $25/20^+$ |
| MULTITASK | GCN | $25/20^+$ | $25/20^+$ | $25/20^+$ | $25/20^+$ |
| MULTITASK | PNA | $15/10^+$ | $15/10^+$ | $15/10^+$ | $15/10^+$ |
| 4,6,8 CYCLES | GIN | $10/10^+$ | $10/10^+$ | $10/10^+$ | $10/10^+$ |
| 4,6,8 CYCLES | GCN | $10/10^+$ | $10/10^+$ | $10/10^+$ | $10/10^+$ |
| 4,6,8 CYCLES | PNA | $10/10^+$ | $10/10^+$ | $10/10^+$ | $10/10^+$ |

Table 5: Dataset statistics. [1]—We used the 10k subset from Dwivedi et al. [32]. [2]—We used the 10k subset from Morris et al. [77]. We generated the datasets from Vignac et al. [98] using an average graph size of [3] 36, [4] 56, [5] 72

| Dataset | # of graphs | # of classes/targets | Average # of vertices | Average # of edges |
|---|---|---|---|---|
| ZINC[1] | 249 456 | 1 | 23.1 | 24.9 |
| ALCHEMY[2] | 202 579 | 12 | 10.1 | 10.4 |
| OGBG-MOLTOX21 | 7 831 | 12 | 18.6 | 19.3 |
| OGBG-MOLTOXCAST | 8 576 | 617 | 18.8 | 19.3 |
| OGBG-MOLFREESOLV | 642 | 1 | 8.7 | 8.4 |
| OGBG-MOLESOL | 1 128 | 1 | 13.3 | 13.7 |
| OGBG-MOLLIPO | 4 200 | 1 | 27.0 | 29.5 |
| OGBG-MOLHIV | 41 127 | 2 | 25.5 | 27.5 |
| OGBG-MOLPCBA | 437 929 | 128 | 26.0 | 28.1 |
| CSL | 150 | 10 | 41.0 | 82.0 |
| MULTITASK | 7 040 | 3 | 18.81 | 47.58 |
| 4 CYCLES[3] | 20 000 | 2 | 36.0 | 30.85 |
| 6 CYCLES[4] | 20 000 | 2 | 48.96 | 43.92 |
| 8 CYCLES[5] | 20 000 | 2 | 61.96 | 56.94 |

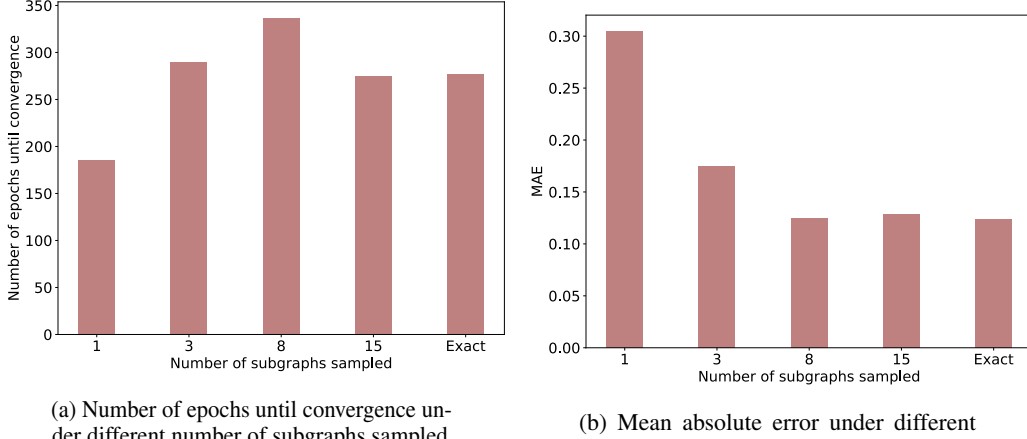

(a) Number of epochs until convergence under different number of subgraphs sampled in training.

(b) Mean absolute error under different number of subgraphs sampled in training.

Figure 3: The impact of sampling subgraphs in $(n-1)$-Reconstruction GIN on the ALCHEMY dataset

# I   Additional results

The attentive reader might wonder how sampling subgraphs impacts the training of the models. More precisely, how does sampling subgraphs affect the convergence and the accuracy of models? To provide insights in this matter, we show results for $(n-1)$-Reconstruction GIN in the ALCHEMY dataset. Figure 3 shows the average number of epochs taken to converge and the training loss at convergence. We note that the training converges faster to a larger loss for a very small sample size, e.g., 1 and 3. Sample sizes of 8 and 15 are already sufficient to converge to approximately the training loss with the exact model. Note that 15 is already the average graph size in the dataset. Thus it has a very similar behavior as the exact model. For a sample size of 8, the model takes 21% more epochs to converge to the same training loss as the exact. Note, however, that using subgraph samples uses a fixed amount of GPU memory independently of the maximum graph size in the dataset.

Table 6: Further results. We highlight in green $k$-Reconstruction GNNs that boost the original GNN architecture. †: Standard deviation not reported in original work.

| | | OGBG-MOLHIV (ROC-AUC %) ↑ | ZINC (MAE) ↓ | ALCHEMY (MAE) ↓ |
|---|---|---|---|---|
| | **GIN** | 75.58 ± 1.40 | 0.278 ± 0.022 | 0.185 ± 0.022 |
| Reconstruction | $(n-1)$ | 76.32 ± 1.40 | 0.209 ± 0.009 | 0.160 ± 0.003 |
| | $(n-2)$ | 77.53 ± 1.59 | 0.324 ± 0.048 | 0.153 ± 0.003 |
| | $(n-3)$ | 75.82 ± 1.65 | 0.329 ± 0.049 | 0.167 ± 0.006 |
| | $\lceil n/2 \rceil$ | 68.43 ± 1.23 | 0.548 ± 0.006 | 0.238 ± 0.011 |
| | **GCN** | 76.06 ± 0.97 | 0.306 ± 0.023 | 0.189 ± 0.004 |
| Reconstruction | $(n-1)$ | 76.83 ± 1.88 | 0.248 ± 0.011 | 0.162 ± 0.002 |
| | $(n-2)$ | 76.13 ± 1.18 | 0.340 ± 0.025 | 0.157 ± 0.004 |
| | $(n-3)$ | 76.00 ± 3.30 | 0.361 ± 0.015 | 0.161 ± 0.004 |
| | $\lceil n/2 \rceil$ | 72.16 ± 1.96 | 0.544 ± 0.006 | 0.236 ± 0.013 |
| | **PNA** | **79.05** ± 1.32 | 0.188 ± 0.004 | 0.176 ± 0.011 |
| Reconstruction | $(n-1)$ | 77.88 ± 1.13 | 0.170 ± 0.006 | 0.125 ± 0.001 |
| | $(n-2)$ | 78.49 ± 1.33 | 0.197 ± 0.007 | 0.128 ± 0.002 |
| | $(n-3)$ | 78.85 ± 0.48 | 0.212 ± 0.212 | 0.152 ± 0.006 |
| | $\lceil n/2 \rceil$ | 76.48 ± 0.35 | 0.582 ± 0.018 | 0.243 ± 0.005 |
| | **LRP** | 77.19 ± 1.40 | 0.223 ± 0.001 | — |
| | **GSN** | 77.99 ± 1.00 | **0.108** ± 0.001 | — |
| | $\delta$-**2-LGNN** | — | 0.306 ± 0.044 | **0.122** ± 0.003 |
| | **SMP** | — | $0.138^\dagger$ | — |

# J   Result Analysis

## J.1   Graph Property Results

Such results are a consequence of the benefits of $k$-Reconstruction GNNs highlighted in Theorem 1. That is, when we remove vertices we transform cycles into paths and trees, graphs easily recognizable by GNNs. By reconstructing the original cycle from its subgraphs, we are able to solve the tasks. Moreover, reconstruction is also able to make GNNs solve the multitask of determining a graph's connectivity, diameter and spectral radius. When we remove a vertex (or a connected subgraph) from a graph, we expect to change only the representation of vertices in its connected component, thus, by inspecting unchanged representations we can determine connectivity and solve the task. The help of reconstruction in solving graph diameter and spectral radius is a bit less direct. The first is related to shortest path lengths and the latter to the number of walks of size $n$ on the graph. When we remove vertices, we alter vertex representations. The way these representations are changed are affected by both shortest path lengths and number of walks. Thus, by measuring the change in representations, reconstruction can better capture these metrics. Finally, we observe that opposed to Ring-GNN, PPGN and Positional GIN, $k$-Reconstruction GNNs is the only method consistently solving the tasks while maintaining the most important feature of graph representations, i.e., invariance to vertex permutations.