# OpenReview forum: "Reconstruction for Powerful Graph Representations"
_NeurIPS.cc/2021/Conference — NeurIPS 2021 Poster_

### Official Review · Reviewer_TP2j · 2021-07-08

**Rating:** 7
**Confidence:** 4

**Summary:**

This paper bridges ideas from graph theory and graph representation learning to propose a simple but theoretically backed-up strategy to increment the expressive power of Deep Graph Networks (DGNs).
The underlying idea should be that, by considering subset of a graph where some vertices are removed, it is possible to discriminate between graphs that would appear identical to the original DGN.


**Limitations And Societal Impact:**

Yes.

**Main Review:**

This is an example of a very good paper. It is well written and organised, the bibliography is wide and complete, and it guides the reader towards the main findings, giving a substantial contribution to the field of graph representation learning. Given the amount of work put in by the authors, most of which was relegated to the appendix, submission to a journal may have been more appropriate.
The synthetic experimental part seems solid, but the real-world experiments suffer from the fact that the hyper-parameters have been fixed in advance. Thus, the comparison is actually unfair; it might be possible that on those real-world datasets a proper model selection over GIN, GCN and PNA could have nullified the reported performance improvements of the proposed approach. While the same issue occurs on synthetic datasets, given the way they are created it is less likely that the main results would have changed. While it would be highly appreciable to see the authors fix this mistake, the overall contribution is still undoubtedly valuable. Below there are some requests of clarifications and potential suggestions (mainly regarding style and presentation) that may help improve the mauscript.

line 11: "any GNNs" is too strong of a claim. Please consider changing the wording.
line 24: claiming that the representational limit of DGNs are upper-bounded by the WL-test may not be appropriate, as WL does not consider vertex/edge attributes (see also line 48-50).
line 37: it may be important to state from the beginning that the "universal approximation" property is related to graph isomorphism without attributes (unless I missed something).
line 60: what do the authors mean by "the combination of theoretical and empirical results is an additional connection"? Is it not enough to have incorporated the theoretical insights into the experiments of Section 5?

line 93-99: for someone that is not familiar with the reconstruction conjecture, this part may be vague. Please consider moving it a bit later, perhaps in a different form after the conjecture has been formulated.
lines 108-109: the term "edge directions" becomes clear later on in the paper (line 139). The authors may consider removing this reference or explaining it.
line 116: Question: if R can distinguish pairs of graphs G,H, is it implied that R distinguishes the individual graphs according to the previous definition? Some clarification may be in order here, perhaps using a more formal notation.

line 143: so far, reconstructible graphs have been defined, wheres reconstructible graph properties have not. What is a reconstructible property (formally)? Is it related to the concept of hereditary property defined later? Please clarify this point.
line 178: it seems the term $k$ is used both as an index taking different values as well as the exact size of the graphs under consideration. Please consider using a different index to help the reader.
line 188: similarly, it would be helpful to explicitly state that r^3 is considered more expressive than r^4 and so on.

line 239: what is the purpose of the ratio |S|\|S_B| ?

lines 251 and 269: it may be confusing that the use of the terms "more expressive" and "less expressive" is in contrasts with the results at lines 263 and 272, respectively. To be more clear, I was expecting a different kind of result after line 269, something of the kind k-Reconstruction "<=" GNN. Perhaps it is just me, but the titles of the paragraph sounded misleading.

line 256: Question: are there any restrictions on the length of the cycle?

line 280: Question: why is it GNN < n-1-reconstruction and not the opposite, given that the authors are conveying the message that n-1-reconstruction is more expressive?
lines 308-309: the hereditary property definition does not necessarily imply that inductive reasoning holds in general. Please clarify this aspect.

line 329: if there is no hierarchy between DGNs and k-reconstruction GNN, why talking about "increased" expressive power in general?

line 397: Question: what is the authors intuition behind the drop in performance of n-3.. n/2 - reconstruction GNNs (which in theory should be more powerful)? Is it because of the sampling approach vs the exact implementation of n-1-reconstruction?

**Time Spent Reviewing:**

4

---

> ### Author Response · Authors · 2021-08-10
> **Reply to Reviewer TP2j**
>
> We thank the reviewer for the valuable comments. In what follows, we discuss and clarify your suggestions.
>
> > *"claiming that the representational limit of DGNs are upper-bounded by the WL-test may not be appropriate, as WL does not consider vertex/edge attributes"*
>
> That is an important point that the earlier literature often omits. Although WL does not support continuous vertex/edge attributes, the WL hierarchy is linked 1-to-1 to hierarchies of universal permutation-equivariant functions over graphs with continuous vertex attributes, see https://arxiv.org/abs/2006.15646 (ICLR 2021).
>
> > *"Question: if R can distinguish pairs of graphs G,H, is it implied that R distinguishes the individual graphs according to the previous definition? Some clarification may be in order here, perhaps using a more formal notation."*
>
> Thanks, we will further clarify that in the final version. $\mathfrak{R}$ distinguishes a graph $G$ if it can distinguish $(G,H)$ **for all** $H$ $\in$ $\mathcal{G}$. Note that $\mathfrak{R}$ can distinguish between $G$ and $H$ while not giving them unique representations.
>
> >*"What is a reconstructible property (formally)? Is it related to the concept of hereditary property defined later? Please clarify this point.*"
>
> We will clarify this point. In graph theory, a graph property is a property that is closed under isomorphism, e.g., planarity or whether it contains a cycle. A graph property can be seen as a binary graph function that outputs 1 if the graph has the property and 0 otherwise. Thus, a reconstructible graph property is simply a binary reconstructible function as defined in Def. 2. A hereditary property is simply a specific type of graph property. We understand the source of confusion and will make this statement clear in the final version.
>
>
> > *"it seems the term  is used both as an index taking different values as well as the exact size of the graphs under consideration. Please consider using a different index to help the reader."*
>
> There may be some misunderstanding here. We will clarify this since l.178 $k$ is not an index but rather a fixed subgraph size.
>
>
> > *"what is the purpose of the ratio $|S|/|S_B|$ ?"*
>
> We will better describe its role. Note that we want to approximate the sum over all subgraphs if we use a Deep Sets function. Thus, the ratio $|\mathcal{S}^{(k)}|/|\mathcal{S}^{(k)}_B|$ makes the expected value of the approximation equal to the sum. If you would like to use a mean aggregation over the subgraphs instead, you can simply average the estimates with the ratio $1/|\mathcal{S}^{(k)}_B|$.
>
> > *"lines 251 and 269: it may be confusing that the use of the terms "more expressive" and "less expressive" is in contrasts with the results at lines 263 and 272, respectively. To be more clear, I was expecting a different kind of result after line 269, something of the kind k-Reconstruction "<=" GNN. Perhaps it is just me, but the titles of the paragraph sounded misleading."*
>
> Yes, we understand the confusion and will clarify it. The title of the paragraphs was the research questions that guided our findings. However, we agree that titles stating the results explicitly can be more helpful, and we will incorporate this change in the final version. Excellent suggestion, thank you!
>
>
> > *"line 256: Question: are there any restrictions on the length of the cycle?"*
>
> No, there are no restrictions. The result stands for any cycle graph (note that cycle graphs have more than 2 vertices).
>
> > *"line 280: Question: why is it GNN < n-1-reconstruction and not the opposite, given that the authors are conveying the message that n-1-reconstruction is more expressive? "*
>
> Please refer to our notation in l.117. It means that GNNs are strictly less expressive than (n-1)-Reconstruction GNNs, as you correctly understood.
>
> >*"lines 308-309: the hereditary property definition does not necessarily imply that inductive reasoning holds in general. Please clarify this aspect."*
>
> We are not quite sure what the reviewer means by “the hereditary property def. does necessarily imply that inductive reasoning holds in general”. If a property is defined as hereditary and no restrictions on the graph size are imposed, we can apply induction to all its subgraphs.
>
> > *"line 329: if there is no hierarchy between DGNs and k-reconstruction GNN, why talking about "increased" expressive power in general?"*
>
> We agree that “change in expressive power” is more suited in a general context (if you do not incorporate the original GNN representation). We will make this change in the final version.
>
> > *"line 397: Question: what is the authors intuition behind the drop in performance of n-3.. n/2 - reconstruction GNNs (which in theory should be more powerful)? Is it because of the sampling approach vs the exact implementation of n-1-reconstruction?"*
>
> Good point we need to clarify. First, we will note that $n/2$ is not more powerful than original GNNs (see Prop. 2). That is, $n-3$ and $n-2$ can have decreased power due to the higher variance in the subgraph sampling procedure (for a fixed number of samples). We will add this observation to the final version.
>
> > *"The synthetic experimental part seems solid, but the real-world experiments suffer from the fact that the hyper-parameters have been fixed in advance. Thus, the comparison is actually unfair; it might be possible that on those real-world datasets a proper model selection over GIN, GCN and PNA could have nullified the reported performance improvements of the proposed approach. "*
>
> We agree that the models can benefit from further hyper-parameter tuning. However, we note that one of our goals was to show that reconstruction can be used as a simple add-on to your favorite GNN, making it solve new tasks and improve its generalization. Finally, by not extensively tuning hyperparameters, it becomes more apparent that the boost in performance comes from graph reconstruction. We will add a discussion and ablation studies with hyperparameter tuning in the final version, which will likely show more benefits for our approach.

---

> > ### Comment · Reviewer_TP2j · 2021-08-27
> > **Many thanks for your response**
> >
> > I am satisfied with the answers provided. Thanks for the explanation. In light of what is written above and the discussion with the reviewers, I will keep my score unchanged.

---

### Official Review · Reviewer_8wLo · 2021-07-16

**Rating:** 6
**Confidence:** 4

**Summary:**

The paper uses principles of the reconstruction conjecture to devise powerful graph representations addressing the current challenges in graph representation learning. The authors first present a theoretical construction of reconstruction-neural-networks, which is impractical because a universal graph function is not learnable. They then construct an approximate k-reconstruction-GNNs, which replace the graph representation function with GNN layers. The authors also discuss the hierarchy of expressive power and the relation of k-reconstruction networks to GNNs expressive power. They show theoretical guarantees regarding the expressive power of the suggested model, observing that under some conditions, k-reconstruction networks can distinguish graphs that GNNs cannot and even 2-GNNs cannot. While under other conditions, the opposite is true. The authors conduct an extensive set of experiments showing the model’s ability to distinguish several types of structures GNNs cannot, and also conduct some experiments on real-world tasks.

**Limitations And Societal Impact:**

No societal impacts discussed

**Main Review:**

1. **Originality**

   The paper provides a novel combination of a classical conjecture from graph theory with graph representation learning. I find the ideas and theory presented in the paper interesting.

2. **Experimental Evaluation**

   - The authors provide a wide dissection of the different configurations of the suggested method illustrating a boost in performance when used with common GNN backbones.
   - The results on real-life dataset are ok. But the graphs the method is applied to are relatively small, which leads to item (3) below stating my concerns.

**Convergence and Generalization**

In the approximation of the k-reconstruction GNNs the authors resort to averaging over subsets of the k-decks due to the complexity reasons. My questions are:

**(a)** When using subsets of the k-decks, although the authors provide a theoretical justification in line 242, how does the subset sampling effect the convergence of the training?

**(b)** Could the proposed method generalize well to large graphs?



---
After reading the author's response and the other reviews, I still lean towards acceptance. Although empirical results are not exceptional, I believe the theoretical concepts and relations to reconstruction conjecture from graph theory are valuable and may lay ground for future advancements in GRL. I hereby keep my score unchanged.

**Time Spent Reviewing:**

5

---

> ### Author Response · Authors · 2021-08-10
> **Reply to Reviewer 8wLo**
>
> We appreciate the useful comments and feedback from the reviewer. Below we address your questions.
>
> > *"When using subsets of the k-decks, although the authors provide a theoretical justification in line 242, how does the subset sampling effect the convergence of the training?"*
>
> We point the reviewer to Table 4 in the supplement, showing which models were trained with subgraph subset sampling. As we can see there, those models converged either to a better or a comparable test performance as the original GNN. We will add a plot with example training curves to the final version. Further, we would also like to note that subgraph sampling did not impact the results in Table 2, which explored the expressive power of Reconstruction GNNs.
>
> > *"Could the proposed method generalize well to large graphs?"*
>
> Possibly. Our work focused on smaller graphs because these would see the largest whole-graph representation changes when removing vertices. We conjecture that the representation of larger graphs would not change substantially (we have preliminary results but not a sufficiently convincing test of this hypothesis). We are looking for graph classification datasets with larger graphs to see if we can get a definitive answer to add to the supplement in the final version. This would not change the conclusions of the paper.

---

> > ### Comment · Reviewer_8wLo · 2021-08-27
> > **A clarification regarding convergence**
> >
> > I thank the authors for their answers.
> >
> > I want to clarify my question regarding the convergence of training. I am aware that the achieved performance is either superior or on par. My intent, however, was, does the sampling slow down the convergence of training due to the stochasticity? If it does, by how much? An interesting ablation would be to use different sizes of subsets.

---

> > > ### Author Response · Authors · 2021-08-28
> > > **Additional experiments regarding convergence**
> > >
> > > Thank you for further clarifying your question. We agree it is an excellent extra question to be added in the final version of the appendix. We will add plots with training curves varying the subgraph minibatch size as you suggested. Unfortunately, the NeurIPS system will not allow us to edit the paper right now.  Still, we generated preliminary results for $(n−1)$-Reconstruction (GIN) in the alchemy dataset for a brief discussion here.
> > >
> > >
> > > The table below shows the average number of epochs taken to converge and the training loss at convergence. We note that the training converges faster to a larger loss for a very small sample size, e.g., 1 and 3. Sample sizes of 8 and 15 are already sufficient to converge to approximately the training loss with the exact model. Note that 15 is already the average graph size in the dataset. Thus it has a very similar behavior as the exact model. For a sample size of 8, the model takes 21% more epochs to converge to the same training loss as the exact. Note, however, that using subgraph samples uses a fixed amount of GPU memory independently of the maximum graph size in the dataset.
> > >
> > > | Dataset | Model                      | # of subgraphs sampled| # of epochs | Training loss |
> > > |---------|----------------------------|---|---|---|
> > > | Alchemy | ($n-1$)-Reconstruction GIN | 1 | 186  |  0.305 |
> > > | Alchemy | ($n-1$)-Reconstruction GIN | 3 |  290 | 0.175  |
> > > | Alchemy | ($n-1$)-Reconstruction GIN | 8 |  337 |  0.125 |
> > > | Alchemy | ($n-1$)-Reconstruction GIN | 15 |  275 | 0.129  |
> > > | Alchemy | ($n-1$)-Reconstruction GIN | Exact | 277  | 0.124  |

---

### Official Review · Reviewer_Dm2T · 2021-07-16

**Rating:** 5
**Confidence:** 3

**Summary:**

This paper proposed several kinds of GNN models that integrate the representations of subgraphs to obtain graph representations.

First, the k-reconstruction NN is defined using the representation of a subgraph of size k for the input graph G. Under the reconstruction conjecture, the k-reconstruction NN is proven to be most expressive.

Next, this paper defined the k-reconstruction GNN, which is made by specializing k-reconstruction NN by obtaining representations using GNNs. The computational complexity of a k-reconstruction GNN is exponential with respect to k. Therefore, we proposed a method to approximate it by sampling subgraphs. The relationship between GNNs and k-reconstruction GNNs is discussed. Concretely, the k-reconstruction GNN is not less expressive than GNN if k is not too small. Also, k-reconstruction GNN is not more expressive than GNNs when k is small.

Finally, the proposed k-reconstruction GNN was applied to synthetic and real datasets (OGB). This paper showed that the proposed model improved the accuracy of the proposed GNN compared to baseline GNNs.

**Ethical Concerns:**

N.A.

**Limitations And Societal Impact:**

- [L1] The authors discussed the problem of computational complexity of k-Reconstruction NN in l.208.
- [L2] The authors showed the theoretical limitation k-Reconstruction GNN in terms of expressive power.


**Main Review:**

### Strength

- [S1] We can apply the k-reconstruction GNN to any GNN for graph prediction tasks (in the experiments, this paper used GCN, GIN, and PNA).
- [S2] The proposed model is provably no worse than GNNs in terms of expressive power.

### Weakness

- [W1] I would say that there is room for improvement in the organization of the paper.
- [W2] Also, I think there is room for improvement in the proof of mathematical statements.
- [W3] Existing research employed the idea of reconstruction theory for the theoretical analysis of structured representation learning. Therefore, I think the novelty of the paper in this respect is limited.

### Soundness (Do theorems and experiments answer research questions, assuming they are correct?)

- [So1] The research question of this paper is to explore the relationship between graph theory and graph representation learning (GRL). More precisely, this paper tried to create a simple, scalable, and expressive GRL model using reconstruction theory. For the following reasons, I think that this paper is an answer to the research question from a theoretical perspective.
  - [So1.1] The proposed k-Reconstruction GNN is not less expressive than the corresponding GNN (Theorem 1, 2).
  - [So1.2] When the true distribution satisfies the hereditary property, the proposed architecture can reduce the variance of the empirical loss (Theorem 3). However, it requires strong convexity assumptions about the loss function and the architecture.
- [So2] As for the experimental side, I have both positive and negative opinions.
  - [So2.1] The proposed model is effective in the synthesis and real datasets of the graph classification task (Table 1, 2).
  - [So2.2] Table 1 shows that vanilla GIN and GCN can recognize cycles to some extent and that the advantage of the reconstruction model is not large.
  - [So2.3] As for GIN and GCN, the proposed model has improved the prediction accuracy on the OGB dataset, demonstrating the empirical effectiveness.

### Correctness (Are derivation of theorems and experiments correct?)

- [C1] I think the statement and proof of the theorem are written informally. Therefore, it was difficult for me to verify its correctness. I would like this paper to define concepts more mathematically rigorously write the statement more formally.
  - [C1.1] For Proposition 1, I tried to give formal proof by myself, but I could not do so, and I could not verify its correctness.
  - [C1.2] Definition 6, I did not understand what was meant by "give a unique representation to a graph sampled with probability 1-ε".
  - [C1.3] In Definition 6, ε-universal is defined for a distribution D on the graph. On the other hand, (ε, k)-universal is defined for the induced subgraph. Is this correct that (ε, k)-universal is defined for the probability distribution on a graph of size k, obtained by sampling the graph from D and then uniformly randomly sampling more subgraphs of size k from it?


**Novelty and Significance (Do the paper have novel points? If so, are they significant?)**

- [N1] When we consider representation learning for graph structures, it is natural to compute representations for subgraphs and aggregate them. The idea of using subgraphs in a graph for GRL has been used in Graph Homomorphism Network (GHN) [Nguyen and Maehara, ICML2020], Graph Substructure Networks (GSN) [19], and Structural Message Passing (SMP) [Vignac et al., NeurIPS2020]. However, this paper has novel points because the proposed method of this paper is different from these works in how we extract subgraph information.
- [N2] As pointed out by this paper, Shawe-Taylor [88] and Bouritsas et al. [19] used reconstruction theory to guarantee expressive power. Therefore, the novelty of the use of reconstruction theory for GRL is limited. On the other hand, this paper is novel because it pointed out the problems of these architectures from the viewpoint of computational complexity and proposes a model with reduced computational complexity.
- [Nguyen & Maehara, ICML2020] http://proceedings.mlr.press/v119/nguyen20c.html
- [Vignac et al., NeurIPS2020] https://proceedings.neurips.cc//paper_files/paper/2020/hash/a32d7eeaae19821fd9ce317f3ce952a7-Abstract.html

**Clarity (Is the paper clearly written?)**

- [Cl1] I would say that there is room for improvement in the clarity of the paper.
- [Cl2] I think there is room for improvement in the proof of mathematical statements.
- [Cl3] In addition, this paper introduced several similar concepts (k-Reconstruction NN, full Reconstruction NN, and Reconstruction GNN). It was difficult for me to grasp the relationship between them. So, I would like this paper to clarify the relationship between them more explicitly.

**Other Comments**

- [O1] In Theorem 1, with respect to which variable the order of $o(1)$ is written?
- [O2] Some model definitions have $n$ (e.g. $(n-1)$-reconstruction GNN). Is it assumed that such models accept only graphs of size $n$ as input?
- [O3] 2-WL is undefined. If it is to be used in mathematical statements, its definition should be clarified.
- [O4] In l.156--164 and Appendix C, existing results on graph reconstruction are presented. It is interesting in itself, but I could not understand how it is related to the model proposed in this paper. For example, does it provide background ideas for the proposed model? Or can we analyze the expressive power of the proposed model by these existing studies?
- [O5] It was difficult to grasp the meaning of l.215--223 when I read it for the first time. This paragraph have what is done in this section and what is done in the Appendix in an interleving manner. It might hindered my understanding.
- [O6] Is it correct that Var in Theorem 3 is the variance with respect to both the probability distribution $P(G, Y)$ and the randomness coming from sampling a subgraph from $G$?
- [O7] In l. 910, I think it is better to write something like $\mu_k(G) = E_{H\sim \mathcal{G}_k(G)}[\mu(H)]$. Also, I think it is better to state explicitly that H is uniformly sampled from $G_k(G$).
- [O8] For Theorem 1, I want to discuss the condition of $\ell$ (or equivalently $k$) in more detail because if I understand correctly, it just borrowed from conditions of the existing studies. Can we think that l should be sufficiently small (specifically, $O(\log n/\log \log n)^{1/2})$)?


### Post-rebuttal comments

My main concern in the initial review comments was about the accuracy of mathematical statements. I asked them to the authors and they answered my questions properly. Taking them into consideration, although I keep my rating (5), I do not object to accepting the paper, providing that the authors improve the mathematical descriptions in the camera-ready version.


**Time Spent Reviewing:**

8

---

> ### Author Response · Authors · 2021-08-10
> **Reply to Reviewer Dm2T**
>
> We thank the reviewer for the useful comments. We want to clarify some important points.
>
> > *"For Proposition 1, I tried to give formal proof by myself, but I could not do so, and I could not verify its correctness."*
>
> Since we can interact on OpenReview, we would gladly help to understand where the reviewer got stuck and why. The proof in the supplement is complete, albeit very compact (which makes it not too reader-friendly). We want to expand it in the final version, adding more details.
>
> The basic idea is that a function is $k$-reconstructible if invariant under $k$-decks, i.e., it assigns the same value to graphs with the same multiset of $k$-size subgraphs. Note that $k$-reconstruction NNs use a multiset of most-expressive representations of subgraphs, i.e., their isomorphism types. Thus, we can think of the input of $k$-reconstruction NNs as the $k$-deck of a graph. If $f_{\mathbf{W}}$ is an universal approximator of multiset functions, it can approximate any function over $k$-decks, i.e. $k$-reconstructible functions.
>
> > *"[C1.2] Definition 6, I did not understand what was meant by "give a unique representation to a graph sampled with probability 1-ε"."*
>
> Let $\mathcal{D}$ be a distribution on graphs. A graph representation is $\varepsilon$-universal with $\varepsilon$ in $[0,1)$ for $\mathcal{D}$ if it assigns, with probability $(1-\varepsilon)$, a unique representation, up to isomorphism, to a graph sampled from $\mathcal{D}$.
>
> We will clarify in the text that the probability is with respect to the sampling from $\mathcal{D}$, the representation itself is not random. $1-\epsilon$ is the probability of sampling a graph that gets a unique representation.
>
> > *"[C1.3] In Definition 6, ε-universal is defined for a distribution D on the graph. On the other hand, (ε, k)-universal is defined for the induced subgraph. Is this correct that (ε, k)-universal is defined for the probability distribution on a graph of size k, obtained by sampling the graph from D and then uniformly randomly sampling more subgraphs of size k from it?"*
>
> Not quite (we will clarify this in the paper, thanks). There is no sampling of subgraphs in this definition. A representation is $(\epsilon,k)$-universal if a graph sampled from $\mathcal{D}$ has all of its non-isomorphic $k$-size subgraphs assigned a unique representation.
>
> > *"[N1] When we consider representation learning for graph structures, it is natural to compute representations for subgraphs and aggregate them. The idea of using subgraphs in a graph for GRL has been used in Graph Homomorphism Network (GHN) [Nguyen and Maehara, ICML2020], Graph Substructure Networks (GSN) [19], and Structural Message Passing (SMP) [Vignac et al., NeurIPS2020]. However, this paper has novel points because the proposed method of this paper is different from these works in how we extract subgraph information."*
>
> We appreciate the input. We have cited Vignac et. al 2020 and will add Nguyen & Maehara 2020 to the final version.
>
> > *"In Theorem 1, with respect to which variable the order of $o(1)$ is written?"*
>
> With respect to $n$ (note that the rest of the expression depends on $n$).
>
> > *"Some model definitions have $n$ (e.g. $n-1$-reconstruction GNN). Is it assumed that such models accept only graphs of size  as input?"*
>
> No, $n$ is arbitrary, not fixed. An ($n-\ell$)-Reconstruction GNN means we remove $\ell$ vertices from the graph independently of its size. That is, the subgraph size adapts to the size of the graph.
>
> > *"2-WL is undefined. If it is to be used in mathematical statements, its definition should be clarified."*
>
> Thanks for the feedback. We used the classical definition of 2-WL (see https://people.cs.umass.edu/~immerman/pub/opt.pdf) and referenced the reader to it, but  we will add a formal definition in the supplement of the final version.
>
> > *"In l.156--164 and Appendix C, existing results on graph reconstruction are presented. It is interesting in itself, but I could not understand how it is related to the model proposed in this paper. For example, does it provide background ideas for the proposed model? Or can we analyze the expressive power of the proposed model by these existing studies?"*
>
>
> Results from ($k$-)reconstruction theory are central to our work. In the case of $k$-reconstruction NNs and full reconstruction NNs, it directly gives you their expressive power. That is, a function can be approximated by _i)_ k-reconstruction NNs if it is k-reconstructible (Prop. 1) _ii)_ full reconstruction NNs if it is reconstructible (Prop. 4). For $k$-reconstruction GNNs, although the analysis is not as direct, it provides tools to prove their expressive power, see proofs of Theorems 1 and 2 and Figure 2 (supplement). Thus, those results shed light on the limits of models based on $k$-reconstruction.
>
> > *"Is it correct that Var in Theorem 3 is the variance with respect to both the probability distribution and the randomness coming from sampling a subgraph from $G$?"*
>
> Not quite. The variance is with respect to the empirical risk of a $k$-Reconstruction GNN computed exactly. If you sample subgraphs, a new source of variance is introduced and whether the total variance is still reduced or not will depend on the sampling method and the graph structure (which would require special (unrealistic) graphs in order to provide useful bounds).
>
> > *"For Theorem 1, I want to discuss the condition of $\ell$ (or equivalently $k$) in more detail because if I understand correctly, it just borrowed from conditions of the existing studies. Can we think that $\ell$ should be sufficiently small (specifically,  $O(\log n/ \log \log n)^{1/2}$)?"*
>
> Your understanding is correct. The smaller the $\ell$ (i.e., the larger the subgraph), the more we can distinguish graphs. Thus, reconstruction studies how small $\ell$ should be to distinguish a certain property. In practice, to recognize most graph properties, we need sufficiently small $\ell$, i.e., very large subgraphs. This implies that models based solely on $k$-reconstruction (e.g. GSN[2], $k$-ary Relational Pooling[1], $k$-reconstruction NNs) require computing isomorphism types (or most-expressive representations) of very large subgraphs to be expressive. This is one of the main motivations for $k$-Reconstruction GNNs, which allow the use of large subgraphs.
>
> [1] Murphy, R., Srinivasan, B., Rao, V., & Ribeiro, B. (2019, May). Relational pooling for graph representations. In International Conference on Machine Learning (pp. 4663-4673). PMLR.
>
> [2] Bouritsas, Giorgos, et al. "Improving graph neural network expressivity via subgraph isomorphism counting." arXiv preprint arXiv:2006.09252 (2020).

---

> > ### Comment · Reviewer_Dm2T · 2021-08-26
> > **Thank you for your response.**
> >
> > I would like to thank the authors for taking my review comments sincerely and responding to them. I would like to clarify again some of the authors' answers.
> >
> > [C1.1] I thank the authors for explaining the intuition of the proof of Proposition 1. I reread it and understand the outline of the proof. However, I intended that the proof of Proposition 1 is not mathematically precise. Because of this, it was difficult for me to confirm the correctness of the points, which we should carefully discuss mathematically. For example, in the proof of the sufficiency part (i.e., if $r_W^{(k)}$ can approximate the target function $f$, then, $f$ is reconstructible), I think the proof did not take into account of approximation. That is, the proof supports the case in which $r_W^{(k)}$ is exactly equal to the target function $f$. However, when $r_W^{(k)}$ just approximates $f$, then $f$ may not be reconstructible even if $r_W^{(k)}$ is. I guess Proposition 1 is correct even for the approximation cases. However, I would say the proof is not sufficient. Another thing is that I was wondering if we do not have to use the fact $h$ is most expressive in the sufficiency part.
> >
> > [C1.2] I thank the authors for further explanation. However, my question remains. If I read the definition literally, it seems we only draw one graph from the representation. If that would be true, the definition would trivially be satisfied because we can assign a unique representation of the sampled graph. Perhaps this sentence is intended to be a different meaning. However, I couldn't figure out what it was. The same is true of [C1.3].
> >
> > [O2] I think using the variable n to name a GNN that can accept graphs with arbitrary size (e.g., (n-2)-Reconstruction GNN) is inappropriate. For example, the same Reconstruction GNN would be called 98-Reconstruction GNN when it accepts a graph of size 100, and 198-Reconstruction GNN when a graph of size 200.
> >
> > [O4] My question was about the role of Lemma 1 and 2 in the context of this paper. From the authors' response, I understand that Lemma 1 and 2 give examples of functions approximating (hence expressible) by k-Reconstruction Neural Networks. I want to confirm if this is the authors' intention.

---

> > > ### Author Response · Authors · 2021-08-28
> > > **Clarification of reviewer's question/comments**
> > >
> > > We thank the reviewer for reading our response and interacting with us. Next, we clarify the reviewers points further.
> > >
> > > > "(...) in the proof of the sufficiency part (i.e., if $r_{\textbf{W}}^{(k)}$ can approximate the target function $f$, then, $f$ is reconstructible), I think the proof did not take into account of approximation."
> > >
> > > We will try to answer as best as possible since we did not understand what the reviewer means by "take into account of approximation". This direction of the proof is quite simple. The input of $r_{\textbf{W}}^{(k)}$ is the $k$-deck $\mathcal{D}_k$ of a graph. If we allow an approximation error of 0, then the representation can only learn $f$ if $\mathcal{D}_k(G) =  \mathcal{D}_k(H)$ implies $f(G) = f(H)$, which is precisely the definition of $k$-reconstructible functions. For an additive error of $\epsilon$, the condition is that $\mathcal{D}_k(G) =  \mathcal{D}_k(H)$ implies $|f(G) - f(H)| < \epsilon$, which is a relaxed version of $k$-reconstructible functions. It is common to write "approximate" when we refer to an error arbitrarily small, i.e., it holds for every $\epsilon > 0$ (e.g., see appendix A in https://arxiv.org/pdf/2002.04025.pdf). Thus, the function class collapses to $k$-reconstructible functions. We will clarify this in the final version. Thank you for the feedback, please let us know if this is still unclear to you.
> > >
> > > > If I read the definition literally, it seems we only draw one graph from the representation. If that would be true, the definition would trivially be satisfied because we can assign a unique representation of the sampled graph. Perhaps this sentence is intended to be a different meaning
> > >
> > > Note that, as we define in section 2 (l.113), the ability of a representation to distinguish graphs is with respect to the space of all graphs. That single graph we are sampling gets a representation. Is it unique with respect to **all other graphs**? Yes, with probability $1-\epsilon$. Please let us know if this is still unclear.
> > >
> > > > I think using the variable n to name a GNN that can accept graphs with arbitrary size (e.g., (n-2)-Reconstruction GNN) is inappropriate. For example, the same Reconstruction GNN would be called 98-Reconstruction GNN when it accepts a graph of size 100, and 198-Reconstruction GNN when a graph of size 200.
> > >
> > > In the naming, $n$ is not a parameter or a variable. It is just the letter $n$ (we will clarify that the parameter of the model is 2, not the graph size). We found it odd to call it just $(-2)$-Reconstruction GNN. The GNN is always called $(n-2)$-Reconstruction GNN.
> > >
> > >
> > > > My question was about the role of Lemma 1 and 2 in the context of this paper. From the authors' response, I understand that Lemma 1 and 2 give examples of functions approximating (hence expressible) by k-Reconstruction Neural Networks. I want to confirm if this is the authors' intention.
> > >
> > > Yes, you are correct. In particular, the example from Lemma 1 has been recently discussed for vanilla GNNs here https://arxiv.org/pdf/2002.04025.pdf,  NeurIPS 2020.

---

> > > > ### Comment · Reviewer_Dm2T · 2021-08-29
> > > > **Meaning of approximation in Proposition 1.**
> > > >
> > > > I thank the authors for the detailed responses.
> > > >
> > > > [C1.1] I am sorry for my vague question. My intension was approximation between the model $r_W^{(k)}$ and the target function $f$. I think the precise statement of Proposition 1 (sufficient part) should be as follows.
> > > >
> > > > > Let $f$ be a function. Suppose for any $\varepsilon > 0$, there exists a parameter $W(\varepsilon)$ (or any parameters that characterize the model) such that $||f - r_{W(\varepsilon)}^{(k)}|| < \varepsilon$, then $f$ is $k$-reconstructible. Here, $||\cdot||$ is some norm (e.g., supnorm).
> > > >
> > > > By definition, $r_{W(\varepsilon)}^{(k)}$ is $k$-reconstructible for any $\varepsilon$. I think we need additional discussions to prove that $f$ is $k$-reconstructible. One way to fill the gap would be to show that the set of $k$-reconstructible functions is closed with respect to the topology induced by the norm $||\cdot||$.
> > > >
> > > > [C1.2] OK. I understand Definition 6 refers to the definition in l.113.
> > > >
> > > > [O2] OK
> > > >
> > > > [O4] OK

---

> > > > > ### Author Response · Authors · 2021-08-30
> > > > > **Response with regard to approximation in Proposition 1.**
> > > > >
> > > > > Thanks for clarifying the question! This is a really great comment. Indeed, we followed the literature w.r.t. formalism, i.e., neither Wagstaff et al. 2019 nor Zaheer et al. 2017 (http://proceedings.mlr.press/v97/wagstaff19a/wagstaff19a.pdf and https://arxiv.org/pdf/1703.06114.pdf) provided a formal statement on the approximation. Since we used their results, we did not either.
> > > > > We agree with the reviewer that it should be made explicit (and, interestingly, it was never made explicit in the literature). The result we use in Wagstaff et al. 2019 relies on Theorem 9 of Zaheer et al. 2017, which relies on the Stone-Weierstrass theorem for necessity, which implies that the approximation is w.r.t. the supremum norm. We will rewrite the theorem statement to make the approximation explicit (which will mostly match what the reviewer suggests) and rewrite the proof to show the approximation chain in the literature (which, we think, would be immensely helpful to others trying to make more formal statements in the future). Thankfully, this modification does not cascade through the paper since the necessity in Proposition 1 is not used in any of our other results (even those directly using Proposition 1 require only sufficiency).

---

### Official Review · Reviewer_UtnL · 2021-07-16

**Rating:** 4
**Confidence:** 5

**Summary:**

This paper introduced graph reconstruction theory to GNN area and use it to improve the expressiveness of GNNs with some formal theoretical proof. The theoretical framework is not tractable but the author proposed using sampling based method and combine with GNN to  approximate k-reconstruction neural network. Some experiments over synthetic datasets and OGB graph datasets show some improvement over existing message passing based network.

**Main Review:**

1. Originality: although the idea is not presented before, the k-reconstruction idea that views graph as a set of k-subgraphs has some connection to k-WL or k-GNN. As we know that each node in k-WL is a k-size super node which corresponds to a k-subgraph. And the k-WL algorithm over these k-size super nodes are permutation invariant which can be looked as implementation of the f_w function used in the paper. I wish the author provide more formal analysis to connect these two type of models, figuring out their difference and similarity. Nevertheless, from my end the presented k-reconstruction GNN is too complicated and is more like a toy for theoretical analysis instead of real-world useable model.

2. Quality: the author provides a great literature review for expressiveness of GNN. The author also introduced the graph reconstruction theory in section 2. For the definition of reconstructible function, I wish the author can provide more insight of why introducing these defnitions in section 2. A trivial example of reconstructible function is a function that always output a constant representation for all graphs. In section 4 the author use these reconstructible function to characterize the expressiveness of k-reconstruction neural network, but before this I don't see any value of introducing these definitions. I suggest the author rewrite the section 2 part with giving some insight of why introducing these definitions from graph reconstruction theory. For section 3, the presented model looks like theoretical toy model to me. This is very similar to the relational pooling paper but I don't get the value of this model . The contribution of relational pooling is more like the universal approximation ability on graphs but this has been studied extensively. For section 4, the author provided several theorems to characterize the expressiveness of the designed k-reconstruction networks, however these theoretical results are not "tight" at all, most of these theorems just tell us that the k-reconstruction is not worse than GNNs. No amazing theoretical result shows that the k-reconstruction network achieves the best expressiveness (higher than k-WL/k-FWL). Also, the improvement on real world datasets are not much noticeable although it does show some ability in synthetic dataset.

3. Clarity: In general the written is ok, but need some improvement to provide more insight in the flow to help reader understand the paper much easier.

4. Significance: my impression is that the paper is more for theoretical analysis instead of real-world usage. I would like to see some more practical model that can make use of the author's analysis. The author did lots of approximations and it's really hard to know how much expressiveness left after these approximations.

**Time Spent Reviewing:**

3

---

> ### Author Response · Authors · 2021-08-10
> **Reply to Reviewer UtnL**
>
> We thank the reviewer for the useful comments. We want to clarify some important points.
>
> > *"(...) although the idea is not presented before, the k-reconstruction idea that views graph as a set of k-subgraphs has some connection to k-WL or k-GNN. (...) I wish the author provide more formal analysis to connect these two type of models, figuring out their difference and similarity."*
>
> The relationship between $k$-WL/$k$-GNNs and $k$-reconstruction is an interesting question we address in our paper. In Obs. 3 (l.742 supplement), we describe that $k$-WL at initialization (without aggregations) limits the power of $k$-reconstruction NNs.
>
> Note that both $k$-reconstruction NNs and $k$-WL/$k$-GNNs use most-expressive representations of subgraphs, i.e., isomorphism types. On the other hand, $k$-Reconstruction GNNs use GNN representations, which does not imply such a direct relationship (see l. 215). Nevertheless, we provide a result connecting $k$-GNNs and $k$-Reconstruction GNNs in Prop. 3: $(n-2)$-Reconstruction GNNs can distinguish graphs that 2-GNNs cannot.
>
> Finally, note that in practice implementing $k$-Reconstruction GNNs is a much simpler task than implementing $k$-GNNs. Instead of constructing a new graph using the "super-nodes," as you pointed out, we remove a couple of vertices from the original graph and apply the GNN.
>
>
> > *"I suggest the author rewrite the section 2 part with giving some insight of why introducing these definitions from graph reconstruction theory.""*
>
> That is a reasonable idea. However, understanding the definition of ($k$-)reconstructible functions is central to our work. Hence, we believe it should be given a prominent spot.
>
> In the case of $k$-reconstruction NNs and full reconstruction NNs, it directly gives you their expressive power. That is, a function can be approximated by
>
> _i)_ $k$-reconstruction NNs if it is k-reconstructible (Prop. 1)
> _ii)_ full reconstruction NNs if it is reconstructible (Prop. 4).
>
> For $k$-reconstruction GNNs, although the analysis is not as direct, it provides tools to prove their expressive power, see proofs of Theorems 1 and 2 and Figure 2 (supplement). In the final version, we will make these connections more explicit.
>
> > *"For section 3, the presented model looks like theoretical toy model to me. This is very similar to the relational pooling paper but I don't get the value of this model . The contribution of relational pooling is more like the universal approximation ability on graphs but this has been studied extensively. "*
>
> There are two models in Section 3: $k$-reconstruction NNs and full reconstruction NNs. We believe the reviewer is referring to the similarity of $k$-reconstruction NNs and $k$-ary Relational Pooling [1].
>
> We can first emphasize that $k$-ary relational pooling is not proposed as a universal approximator. $k$-ary relational pooling applies relational pooling over $k$-size subgraphs.
>
> Obs. 2 (supplement) shows that $k$-ary relational pooling is a specific type of a $k$-reconstruction NN where a relational pooling procedure gives the subgraph representation. **Why is this connection important?** In the original paper, the authors did not characterize the expressive power of $k$-ary relational pooling. Our results (Prop. 1 and Obs. 2) characterize the power of $k$-ary relational pooling: a function can be approximated by it iff it is $k$-reconstructible.
>
> > *"The author did lots of approximations and it's really hard to know how much expressiveness left after these approximations."*
>
> Approximations are often a necessary evil in graph representation learning methods. Practical GNN expressiveness is also an approximation of the power of the 1-WL algorithm. Table 2 shows how we can solve several tasks originally not solvable by GNNs by applying $k$-reconstruction. Most of the reconstruction models are trained using our approximation procedure (see Table 4 in the supplement for details) and can still solve the tasks. We will better emphasize this in the final version.
>
> > *"Significance: my impression is that the paper is more for theoretical analysis instead of real-world usage. I would like to see some more practical model that can make use of the author's analysis."*
>
> Our theoretical analysis is important in practice (it is not theory for the sake of having theory). For instance, Theorem 3 and Corollary 1 provide conditions in which our method provably has better generalization than GNNs. Such results translate into our real-world experiments. For instance, see ZINC and Alchemy in Table 6 of the supplement.
>
> [1] Murphy, R., Srinivasan, B., Rao, V., & Ribeiro, B. (2019, May). Relational pooling for graph representations. In International Conference on Machine Learning (pp. 4663-4673). PMLR.

---

> ### Comment · Senior_Area_Chairs · 2021-08-27
> **URGENT: reply to the authors**
>
> Dear Reviewer,
>
> Please see below -- your AC has requested that you respond to the author's response.
> This is a crucial step in the decision process that needs your immediate attention.
>
> Thanks,
> SAC

---

### Decision · Program_Chairs · 2021-09-27

**Decision:**

Accept (Poster)

**Comment:**

The reviewers agreed that this work provides a valuable contribution to the GNN research by expanding on the known connections between graph reconstruction theory with graph representation learning. While sometimes falling behind more powerful architectures, the proposed method is shown to consistently improve upon vanilla GNN.

The reviewers originally expressed some concerns about the rigorousness of the mathematical statements. However, these were addressed during the rebuttal phase.

I thus recommend the (revised) paper for publication.